# Bright Tm$^{3+}$-based downshifting luminescence nanoprobe operating around 1800 nm for NIR-IIb and c bioimaging

Yulei Chang [1] ✉, Haoren Chen[1], Xiaoyu Xie[1], Yong Wan[1], Qiqing Li[1], Fengxia Wu[1,2], Run Yang[1], Wang Wang[1] & Xianggui Kong[1]

Fluorescence bioimaging based on rare-earth-doped nanocrystals (RENCs) in the shortwave infrared (SWIR, 1000–3000 nm) region has aroused intense interest due to deeper penetration depth and clarity. However, their downshifting emission rarely shows sufficient brightness beyond 1600 nm, especially in NIR-IIc. Here, we present a class of thulium (Tm) self-sensitized RENC fluorescence probes that exhibit bright downshifting luminescence at 1600–2100 nm (NIR-IIb/c) for in vivo bioimaging. An inert shell coating minimizes surface quenching and combines strong cross-relaxation, allowing LiTmF$_4$@LiYF$_4$ NPs to emit these intense downshifting emissions by absorbing NIR photons at 800 nm (large Stokes shift ~1000 nm with a absolute quantum yield of ~14.16%) or 1208 nm (NIR-II$_{in}$ and NIR-II$_{out}$). Furthermore, doping with Er$^{3+}$ for energy trapping achieves four-wavelength NIR irradiation and bright NIR-IIb/c emission. Our results show that Tm-based NPs, as NIR-IIb/c nanoprobes with high signal-to-background ratio and clarity, open new opportunities for future applications and translation into diverse fields.

Shortwave infrared (SWIR, 1000–3000 nm), also defined as the second near-infrared region (NIR-II), holds great promise for deep tissue imaging in physiological studies and biomedical applications[1–3]. Within this region, emission beyond 1500 nm has been confirmed as another promising imaging window with better spatial resolution and deeper imaging depth[4,5]. Notably, the long ends of the NIR-IIb (1600–1700 nm), NIR-IIc (1700–2000 nm), and NIR-IId/NIR-III (2100–2300 nm) bands show low scattering loss and near-zero autofluorescence, which further enhances spatial resolution, signal-to-background ratio (SBR), and imaging penetration depth to achieve better clarity for bioimaging[6,7]. In particular, this will provide a wider spectral range for multiple imaging without interference from spectral overlap. However, to date, NIR-IIb fluorescent/luminescence probes emitting beyond 1600 nm are still very limited and, to our knowledge, have only been reported for AIEgens with tails extended to 1600 nm[8] and lead sulfide (PbS)/cadmium sulfide (CdS) QDs[6,9–11]. Clearly, more

and brighter probes with high efficiency, low cytotoxicity, large Stokes shift, and photostability are required for these crucial wavelength regions.

Rare earth-doped nanocrystals (RENCs) have become ideal NIR-II imaging nanoprobes with great potential for use in multiplexed sensing[12], time-gated detection[13], imaging-guided therapy[14] and surgery[15] due to their NIR excitation bands, narrow emission bands, and other required advantages mentioned above[16,17]. Notably, the dense energy levels of RE$^{3+}$ cause their emissions to span from the UV–visible to IR spectral regions[18,19]. Several ions have been reported to emit in the NIR-II region, including Ho$^{3+}$ at 1185 nm, Nd$^{3+}$ at 1060 nm, 1310 nm, Er$^{3+}$ at 1525 nm, and Tm$^{3+}$ at 1450 nm[20] (Fig. 1a). It should be noted that Tm$^{3+}$ is one of the few RE$^{3+}$ ions that exhibit different spectral conversions of interest ranging from the UV[21,22] to the mid-infrared regions[23,24]. The commonly reported Stokes or downshifting luminescence (DSL) of Tm$^{3+}$ in the NIR-II region is located at 1450 nm

[1]State Key Laboratory of Luminescence and Applications, Changchun Institute of Optics, Fine Mechanics and Physics, Chinese Academy of Sciences, Changchun 130033, China. [2]Institute of Molecular Medicine, College of Life and Health Sciences, Northeastern University, Shenyang, Liaoning 110000, China. ✉e-mail: yuleichang@ciomp.ac.cn

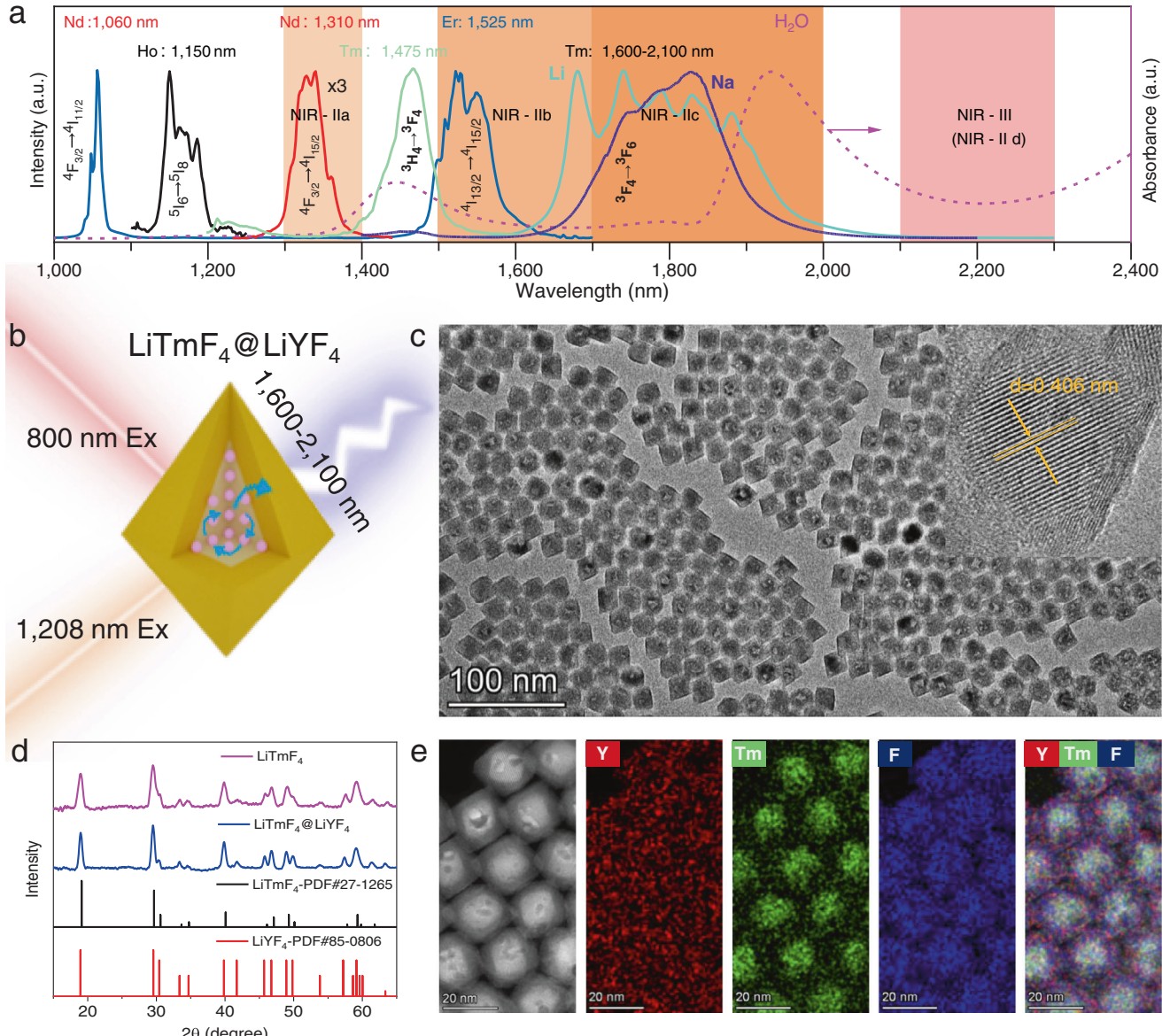

**Fig. 1 | Characterization of Tm-NPs and typical NIR emission of RE³⁺. a** Typical DSL spectra of rare-earth ions (Nd: NaGdF₄:30%Nd; Ho: NaYF₄:10%Yb:1%Ho; Er: NaErF₄@NaYF₄; Tm: NaYF₄@NaYbF₄:0.5%Tm@NaYF₄-powder for 1475 nm and LiYF₄:Tm@LiYF₄-powder for 1600–2100 nm) and absorption spectrum of water in the NIR-II and NIR-III regions. **b** Schematic drawing showing the layout of core-shell structured Tm-NPs. **c** Typical TEM images of the LiTmF₄@LiYF₄ core-shell NPs. Inset: corresponding high-resolution TEM image. **d** XRD patterns for Tm-NPs and standard patterns for LiTmF₄ and LiYF₄. **e** HAADF STEM image and elemental maps of Tm-NPs.

($^3H_4 \rightarrow {}^3F_4$, a weak transition), and it shows a low luminescence efficiency and competes with strong water absorption in this region (Fig. 1a)[2]. Therefore, this dramatically limits the Tm-based probe for optical imaging in the NIR-II region. Nevertheless, in principle, sufficiently strong emission should also bring better image definition because the definition is mainly affected by scattering and background fluorescence. Fortunately, the $^3F_4 \rightarrow {}^3F_6$ transition of Tm³⁺ displays intense emission bands ranging from 1600 to 2100 nm, and high contrast deep tissue imaging could be enabled by further increasing the penetration depth to sub-centimetre levels and eliminating autofluorescence (Fig. 1a). Notably, the detection regions 1700 nm to 1880 nm display absorption and scattering properties similar to those in NIR-IIb[7], recently, Dai et al. extending this range to 2000 nm;[11] thus, it is defined as the NIR-IIc window and is no longer limited to 1700 nm in the NIR-II window (limited by the classic InGaAs detector). To date, this spectral characteristic of Tm³⁺ (1600–2100 nm) is mainly used as a gain material

for laser applications with bulk laser glasses[23,25]. However, there is no report of using Tm³⁺ emission in the NIR-IIb and c regions for in vivo bioimaging with colloidal systems. This is probably due to severe concentration- and surface quenching, inappropriate sensitizer in the preparation of bright Tm-doped nanoparticles.

Herein, to meet the requirement for NIR-IIb- or NIR-IIc-emitting probes, we present for the first time highly doped Tm³⁺ ion self-sensitized nanoparticles for in vivo bioimaging. In this simple design, the Tm³⁺ ions serve as both sensitizers and activators to absorb pump photons at 800 nm (NIR-I) or 1208 nm (NIR-II), and they produce efficient 1600–2100 nm Stokes emission ($^3F_4 \rightarrow {}^3H_6$) due to intense cross-relaxation (CR) caused by this radiative transition. In addition, energy trap ions, for instance, Er³⁺ and Dy³⁺, were rationally doped to mediate Tm³⁺-sensitized downshifting emission. These bright Tm-based NPs enable noninvasive deep-tissue imaging for studies of biological interactions with better quality and clarity in the SWIR window.

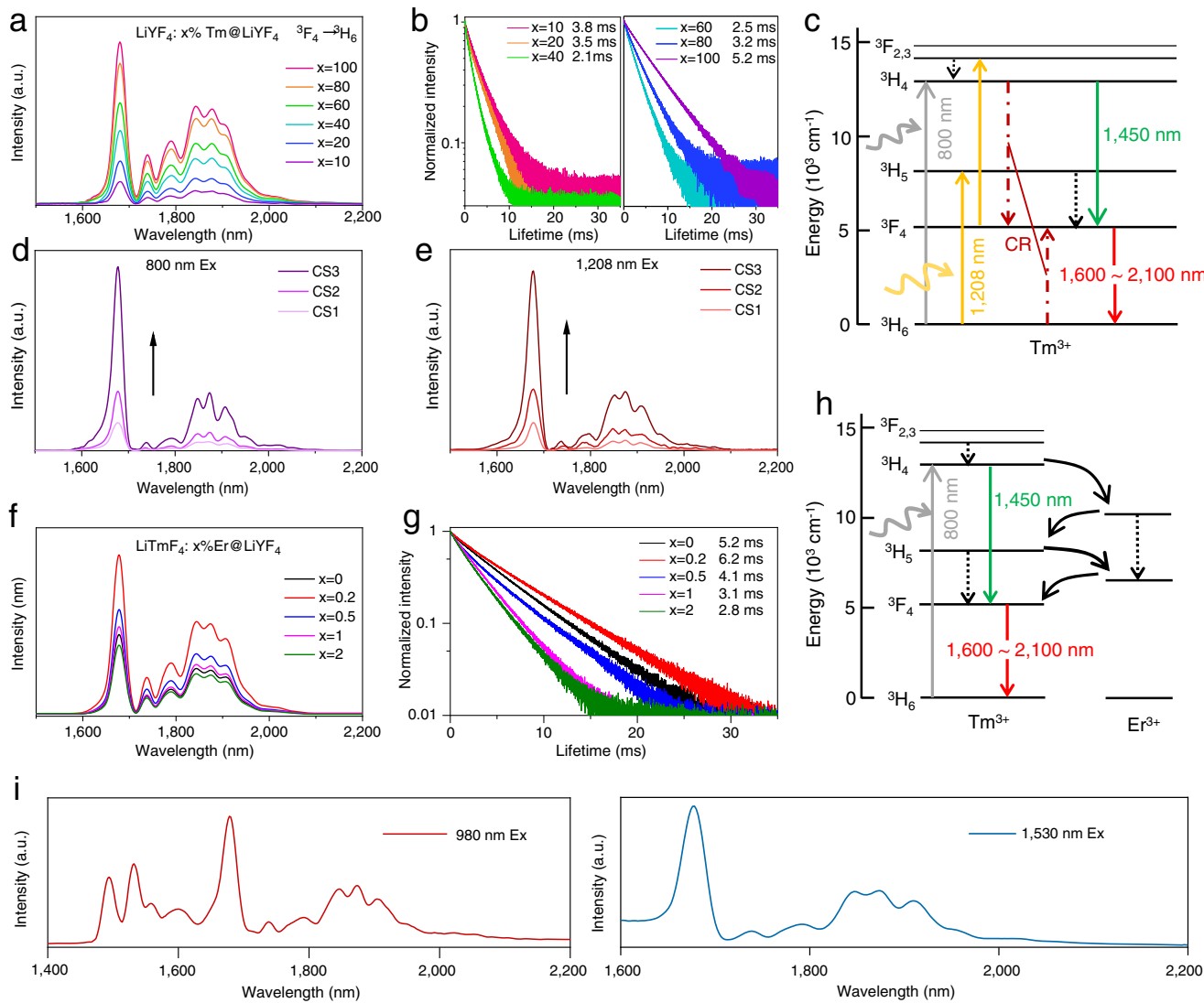

**Fig. 2 | Photoluminescence characterization of the Tm-NPs with enhanced NIR-IIb/c luminescence. a** Emission spectra of the core-shell Tm-NPs with variable Tm³⁺ dopant concentrations in the core showing DSL from 1600 to 2100 nm upon excitation at 800 nm. **b** Corresponding decay times. **c** Proposed downshifting mechanism for Tm³⁺ in the NIR-II region. **d, e** LiTmF₄@Y NPs with different shell thicknesses irradiated at 800 nm/1208 nm. **f** Emission spectra were obtained with various Er³⁺ ions doped into the LiTmF₄: x%Er@ LiYF₄ NPs. **g** Corresponding decay curves for Tm³⁺ emission at 1680 nm. **h** Proposed downshifting mechanisms for LiTmF₄:x%Er@LiYF₄ NPs under 800 nm light excitation. **i** Emission spectra of Tm/Er@Y alloy NPs upon irradiation at 980 nm and 1530 nm.

## Results

Prior to bioimaging applications, we systematically investigated and compared the effects of inert/undoped shells on the luminescence of heavily Tm³⁺-doped nanoparticles. The transmission electron microscopy (TEM) images in Supplementary Fig. 1a and Fig. 1c show that the as-obtained LiTmF₄ core-only and Tm@Y core-shell NPs (Tm-NPs) had uniform morphologies, with average sizes of ~9.8 nm × 13.7 nm and ~19.3 nm × 24.8 nm (width×height), respectively. Their roughly rhomboidal morphologies indicated a tetragonal (I4₁/a) space group, as demonstrated by a powder X-ray diffraction analysis (XRD). The diffraction peaks for Tm-NPs corresponded well to the reference pattern for tetragonal LiYF₄ crystals, indicating the formation of pure phase nanocrystals (Fig. 1d). The HAADF STEM image and element mapping results for the Tm-NPs further confirmed their core-shell structure with a clear component boundary giving rise to a bright Tm³⁺-core and a dark Y³⁺ shell (Fig. 1e). Additionally, compositional analyses with energy-dispersive X-ray spectroscopy confirmed the presence of Li⁺, F⁻, Y³⁺ and Tm³⁺ ions in core-shell Tm-NPs (Supplementary Fig. 2).

We first explored the luminescence properties of single Tm-doped colloidal nanocrystals. DSL of Tm-NPs in cyclohexane was found with SWIR-emission ranging from 1600 nm to 2100 nm, attributed to ³F₄→³H₆ transitions upon 800 nm light irradiation. Due to Stark-splitting substrates with the ³F₄ state, the multiband emission spectra at ~1800 nm were located in the NIR-IIb/c regions, which is suitable for NIR-II imaging. However, compared to the full spectrum of dried Tm-NPs (Fig. 1a and Supplementary 3a), cyclohexane as a solvent severely quenches the luminescence around 1680–1900 nm than other bands due to strong NIR-IIc absorption of cyclohexane (Supplementary Fig. 3b).

For the concentration-mediated DSL of Tm³⁺, severe concentration quenching also occurs in the nanomaterials (core-only, Supplementary Fig. 4), which is consistent with the behavior of the widely reported single-doped bulk materials[26]. Fortunately, after epitaxial growth of the LiYF₄ shell, a steady increase in the DSL intensity was observed (Fig. 2a), and the corresponding TEM images were provided in Supplementary Fig. 5.

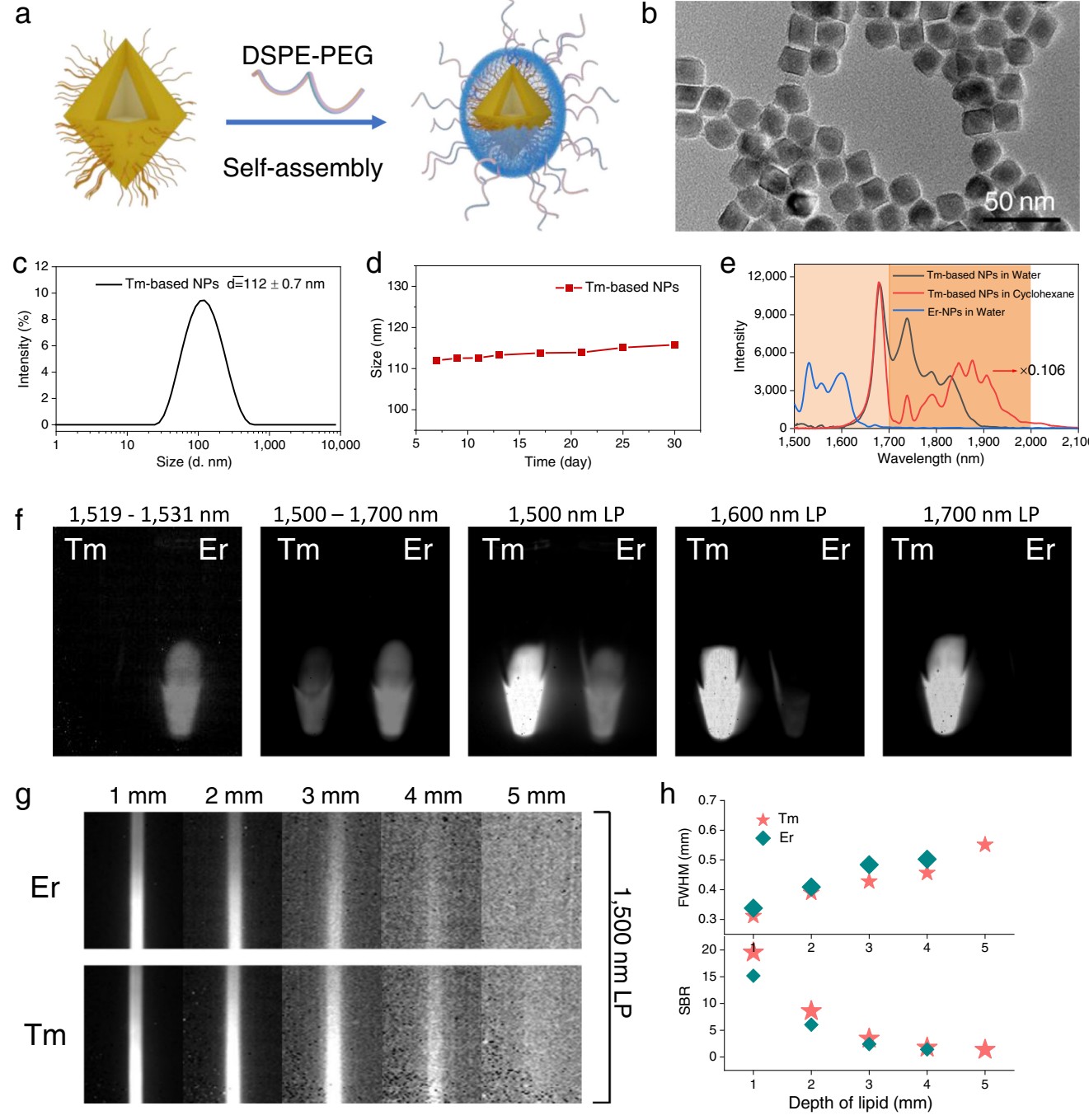

**Fig. 3 | Biocompatibility modification of the Tm-based NPs. a** PEGylation of Tm-based NPs via hydrophobic-hydrophobic interactions between DSPE-PEG and OA-capped Tm(02Er)-NPs. Their corresponding **b** TEM image **c** DLS spectrum after PEGylation and **d** stability of PEGylation Tm-based NPs monitored by DLS. **e** DSL spectra of OA-capped Tm(02Er)-NPs in cyclohexane, Tm(02Er)-NPs@PEG and Er-NPs@PEG in water upon excitation at 800 nm. **f** Tm(02Er)-NPs@PEG and Er-NPs@PEG nanoparticles in vitro under 800 nm excitation (50 mW/cm²) with different filters (corresponding transmission profiles were provided in Supplementary Fig. 17). **g** Luminescence images of a capillary tube filled with Tm(02Er)-NPs@PEG and Er-NPs@PEG solution, immersed in various thicknesses of 1% intralipid solution recorded Er³⁺ or Tm³⁺ emissions beyond 1500 nm upon 800 nm excitation (50 mW/cm²), respectively. **h** Corresponding full width at half maximum (FWHM) and SBR luminescence intensity profiles. Exposure time: 50 ms.

The electronic populating processes can explain the potent concentration-dependent luminescence of Tm-NPs, as shown in Fig. 2c. Tm³⁺ ions were sensitized to absorb 800 nm photons directly by exciting electrons to populate the ³H₄ state through two continuous radiative transition pathways from ³H₄ to ³F₄ (1450 nm) and then from the ³F₄ to ³H₆ (1600–2100 nm). The energy gap between the ³H₄ and the ³F₄ energy level and ³F₄ to the ground state is approximately 6850 cm⁻¹ and 5890 cm⁻¹, respectively. These two close energy gaps

enable efficient Tm³⁺→Tm³⁺ CR between the ³H₄→³F₄ and ³H₆→³F₄ transitions[27,28], which guarantees more absorbed photons populate the intermediate ³F₄ level and then transition to the ground state (³H₆) to increase the intensity of the NIR-II emission. Although luminescence from the ³F₄→³H₆ transition increased monotonically, the decay times of Tm-NPs decreased gradually as the Tm³⁺-doping concentration was increased from 10 to 40%, with further increases in doping concentration from 60 to 100% Tm³⁺, the lifetime for emission at 1680 nm

increased likely due to the size difference (Fig. 2b and Supplementary Fig. 5). The concentration dependence of the competition for the population of the $^3H_4$ and $^3F_4$ energy levels is further supported by the lifetime of the 1450 nm emission. Supplementary Figs. 6a, b show a similar trend for the 1600–2100 nm emission decay time, implying the CR between $^3F_4$ and $^3H_4$ levels. Moreover, the intensity of the 1450 nm emission band was slightly enhanced when the doping level was below 40% due to the increase in the population of the $^3H_4$ state (Supplementary Fig. 6c). As expected, the luminescence intensity was further weakened with high $Tm^{3+}$ concentration because the gradually increased CR inhibited spontaneous emission of the $^3H_4$ state and simultaneously increased the population of $^3F_4$. Furthermore, upon increasing the shell thickness of $LiYF_4$ from ~1.7, 2.8 to 5.0 nm (Supplementary Fig. 1), as expected, emission increased dramatically upon excitation at 800 nm (Fig. 2d), indicating that the epitaxial inert shell separated from the $Tm^{3+}$ activators coupled to the surface[29]. The luminescence lifetimes for emission at 1680 nm changed from 0.2 ms and 0.8 ms to 5.2 ms, indicating that the surface quenching is reduced by thickening the $LiYF_4$ shell and lifetime is particularly sensitive to the shell thickness in the range of 2.8–5.0 nm (Supplementary Fig. 7). Given these, we speculate this phenomenon can be explained in two ways: (1) the inert shell inhibited concentration and surface quenching[29,30], and (2) the increase in the $Tm^{3+}$ concentration enhanced the CR process[28].

Notably, the high concentration of the $Tm^{3+}$ dopant resulted in NIR-II region (1208 nm, $^3H_5$ state) excitation of the Tm-NPs, corresponding absorption spectra as shown in Supplementary Fig. 8a (absorption cross-section: $8.25 \times 10^{-22}$ cm$^2$ at 800 nm and $1.5 \times 10^{-21}$ cm$^2$ at 1208 nm)[9]. Irradiation of these Tm-NPs at 1208 nm resulted in emission peaks and branching ratios identical to those seen with 800 nm excitation. Enhanced DSL emissions with the same trend were observed as the $Tm^{3+}$ ion concentration was increased or the shell thickness was varied (Fig. 2e and Supplementary Fig. 8b), respectively, indicating that these emission bands came from the same transition process. To our knowledge, there is no report in which 1600–2100 nm emission from $Tm^{3+}$ in dispersible NPs was observed, particularly upon excitation at 800 nm and 1208 nm.

Next, we chose dopant ions with energy levels matching those of $Tm^{3+}$, such as $Er^{3+}$ and $Dy^{3+}$, for energy trapping designed to provide brighter NIR-II emissions[31]. We first investigated the $Er^{3+}$ ion dopant with excitation at 800 nm. It is important to note that downshifting emission at 1680 nm was boosted in the presence of $Er^{3+}$ with doping level (<1%, 0.2% is optimal, Tm(02Er)-NPs), and doping with much more $Er^{3+}$ induced a decline trend (Fig. 2f and Supplementary Fig. 9). Furthermore, we investigated corresponding time-resolved populations in the $^3F_4$ state (Fig. 2g). Depopulation of the $^3F_4$ state was accelerated by inhibiting CR of $Tm^{3+}$ through energy transfer between $Tm^{3+}$ and $Er^{3+}$ (Fig. 2h). Interestingly, although the emission intensity at 1600–2100 nm decreased when the doping concentration of $Er^{3+}$ was further increased to more than 10% (Supplementary Fig. 10), the resulting Tm/Er "alloy" NPs (e.g., 30%Er doping, Tm(30Er)-NPs) could even be excited by four-wavelength irradiation in the NIR region, and addition emission bands were observed upon excitation at 980 nm and 1530 nm, respectively (Fig. 2i and Supplementary Fig. 8a, absorption cross-section: $9.5 \times 10^{-22}$ cm$^2$ at 800 nm, $1.075 \times 10^{-21}$ cm$^2$ at 980 nm, $1.75 \times 10^{-21}$ cm$^2$ at 1208 nm and $2.12 \times 10^{-21}$ cm$^2$ at 1530 nm). Notably, due to the increase of population of $^4I_{13/2}$ level of $Er^{3+}$, 1525 nm emission ($^4I_{13/2} \rightarrow {}^4I_{15/2}$) also occurred, such as with 980 nm irradiation. Furthermore, the luminescence quantum yield (1600–2050 nm) was measured to be -14.16% for Tm-NPs and -16.13% for Tm(02Er) NPs with 800 nm excitation (4 W/cm$^2$). These "alloy" NPs open new routes for imaging, anti-counterfeiting, etc., applications involving regulation with multiwavelength selective excitation. However, doping with $Dy^{3+}$ did not boost the NIR emission, which would be strongly quenched by doping a small amount of $Dy^{3+}$ (0.2%) into the $Tm^{3+}$ lattice (Supplementary Figs. 11 and 12) due to the dense ladder-like energy levels of $Dy^{3+}$.

In addition, we optimized this bright emission by the types of heterogeneous epitaxial shells. Comparing the luminescence intensities among Y, Lu, and Gd shells, the $LiYF_4$ shell emits the strongest DSL (see Supplementary Figs. 13 and 14 for details). Next, we further investigated the upconversion behaviors of Tm-NPs. Predominantly monochromatic weak red emission was observed at 696 nm (Supplementary Fig. 15). Furthermore, we extended this work to include the much-studied Na-based counterparts with similar structure and size (Supplementary Fig. 16), e.g., $NaTmF_4@NaYF_4$ and traditional $NaYF_4$:20%Yb,1%Tm@$NaYF_4$ NPs. The former also exhibited luminescence properties comparable with the $LiREF_4$ host under 800 nm excitation but weaker around 1680 nm. Compared to the codoped Yb/Tm system under 980 nm excitation with the same power, Li-based Tm-NPs even exceeded over an order of magnitude that of counterparts. These results further demonstrated the advantage of Li-host in the DSL of $Tm^{3+}$.

## Surface modification of Tm- based NPs for biocompatibility

For bioimaging, the optimized Tm(02Er)-NPs were selected and modified by DSPE-PEG$_{2,000}$ through hydrophobic-hydrophobic interactions (Fig. 3a). After PEGylation, the resulting Tm(02Er)-NPs@PEG was achieved without obvious aggregation, as evidenced by the TEM image in Fig. 3b, and the hydrodynamic diameter determined with dynamic light scattering was 112 ± 0.7 nm (Fig. 3c). It was also used to monitor the long-term colloid stability for 1 month. As illustrated in Fig. 3d, there was no significant change in size, and no dissociation or precipitation occurred during this time, indicating that the Tm-based probes were well dispersed and stable in an aqueous solution. The luminescence intensity of Tm-based probes in aqueous solution decreased much more than it did in cyclohexane (9.4-fold, calculated at 1680 nm) due to the nonradiative deactivation by O−H vibrations (Fig. 3e)[32]. However, the ratio of the DSL intensity at 1738 nm to that of 1680 nm emission increased from 0.37 (cyclohexane) to 0.76 (water) because of the lower absorption of water than that of cyclohexane in this region (Fig. 1a and Supplementary Fig. 3b).

Notably, the DSL of $Tm^{3+}$ can span from NIR-IIb to NIR-IIc. Er-based NPs with emission in the NIR-IIb region (1525 nm) were used as references to quantify the advantages of NIR-IIc emission. Particularly, the highly doped single $Er^{3+}$ core/shell NPs have been recently developed to be high-efficiency NIR-IIb probe[33]. Thus, $LiErF_4@LiYF_4$ NPs with the same Li-based host and shell thickness (5 nm) were used for a relatively fair comparison (Supplementary Fig. 18). Comparable emission intensities of Er- and Tm-based NPs with the same concentration at NIR-IIb and NIR-IIc, were recorded upon 800 nm excitation (Fig. 3e) and were subsequently imaged using a HgCdTe (MCT) camera in separate subwindows (Fig. 3f). The two probes could be clearly distinguished at 1524–1536 nm and >1700 nm range, respectively. Further, even though both could be imaged at 1500–1700 nm sub-window, Er-NPs@PEG gradually became undetectable with the long pass filters changing from 1500, 1600, to 1700 nm. However, the bright Tm-based probe remained. These results indicated that Tm-based NPs would serve as excellent NIR-IIb/c candidates. Moreover, the luminescence images of capillaries filled with these nanoprobes at the same intensities revealed that the emission of Tm-based NPs could still be observed when covered with a 5 mm thick 1% intralipid solution with an SBR of 1.35 and FWHM = 0.55 mm, while that of Er-NPs@PEG was almost undetectable, suggesting that there was considerable penetration as well as clarity on using Tm-based NPs in the NIR-IIb/c region (Fig. 3g, h). Further, 24 h CCK8 assays revealed low cytotoxicity of Tm-based nanoprobe, even at concentrations as high as 800 μg/mL (>90% viability, Supplementary Fig. 19).

## In vivo imaging

To evaluate the feasibility of imaging with the Tm-based probe and realising the desirable clarity and deep tissue penetration in NIR-IIb/c,

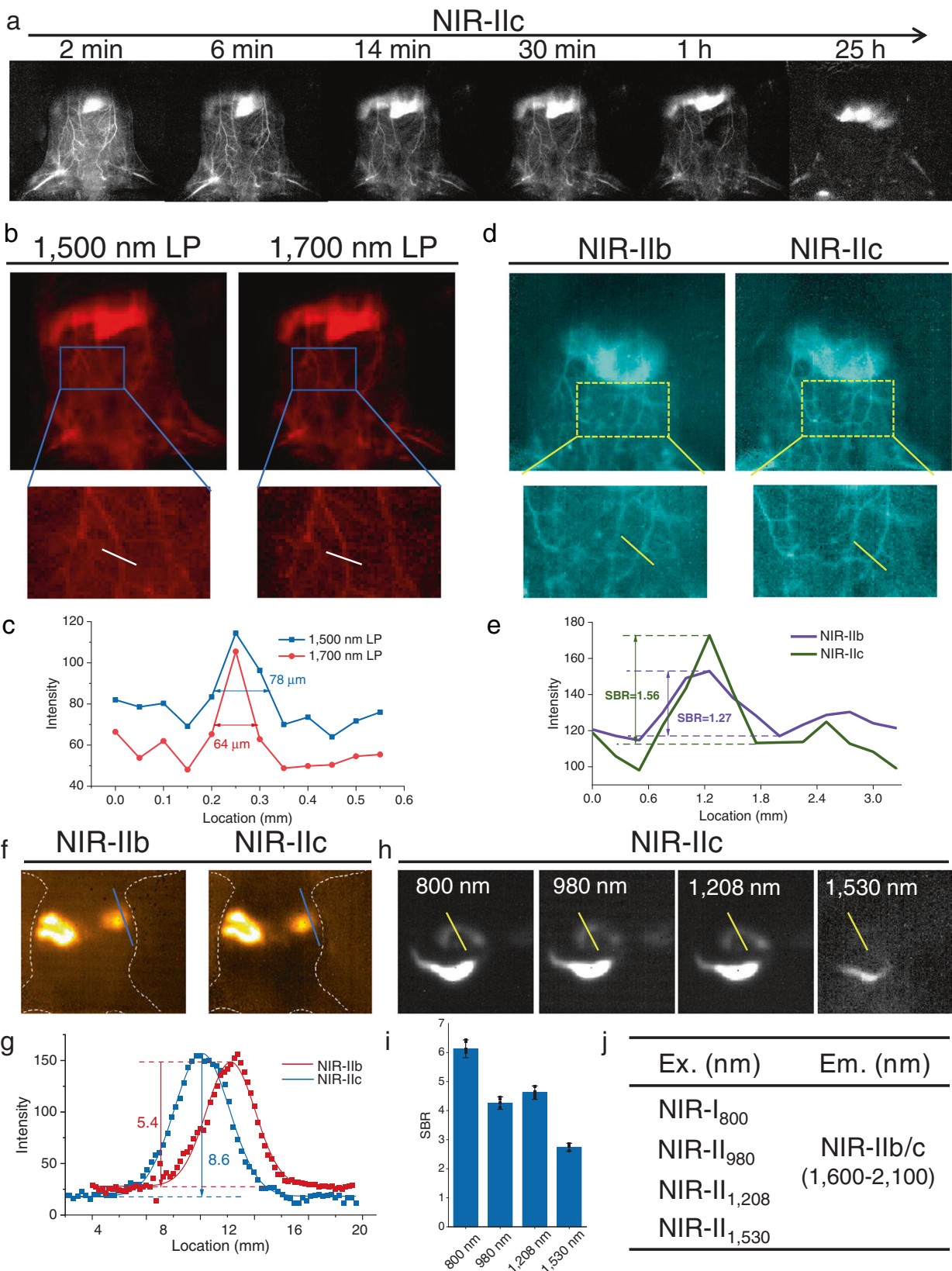

whole body vascular imaging and blood circulation in mice was first demonstrated by detecting luminescence beyond 1700 nm using the MCT camera upon 800 nm excitation. An intense NIR-IIc signal allowed clear visualization of the circulatory system 2 min post-injection. The signal decreased with an increase in post-injection time. The liver was

clearly seen over time (Fig. 4a), indicating that Tm-probe metabolized mainly through hepatobiliary system. Furthermore, the NIR-IIc signal intensity determined by the same vessel exhibited a moderate half-time in the blood (49 min, Supplementary Fig. 20), enabling delineation of vascular hemodynamics under pathological states. Notably,

**Fig. 4 | Real-time and multiwavelength irradiation NIR-IIc imaging demonstrated in vivo.** Real-time vascular luminescence imaging with Tm(02Er)-NPs@PEG in living mice upon 800 nm irradiation (100 mW/cm²) **a** at NIR-IIc region. **b** Vascular imaging at 40 min postinjection points with 1500 nm LP and 1700 nm LP filters, respectively. **c** corresponding luminescence intensity statistics of **b. d**, Vascular luminescence imaging with Tm- and Er-based NPs at the same concentration in NIR-IIb (1500 nm LP + 1700 nm SP filters, 1500–1700 nm) and NIR-IIc (1700 nm LP filter) subwindow. **e** corresponding luminescence intensity statistics of **e. f** Nanoprobe with a mixture of Er- and Tm-based probes at the same intensity in NIR-IIb and NIR-IIc imaging. **g** corresponding SBR statistics of **f. h** GI tract imaging in NIR-IIc (1750–2250 nm) region in living mice gavaged with Tm(30Er)-NPs@PEG upon 800 nm, 980 nm, 1208 nm, and 1530 nm irradiation (120 mW/cm² for each), respectively. **i** Corresponding SBR statistics of **h** data are presented as mean ± s.d. (n = 4). **j** The compiled luminescence profiles for the multiple excitations of Tm/Er-NPs. Exposure time: 100 ms.

the width of the vessel was determined to be 64 μm from the images in the NIR-IIc region (Fig. 4b) by measuring the FWHM through a Gaussian fit compared with that beyond 1500 nm imaging (FWHM = 78 μm), indicating increased spatial resolution in NIR-IIc imaging. These results confirmed the high-resolution imaging performance in the NIR-IIc region. Additionally, vascular imaging using MCT and InGaAs (SD 640) camera was performed to confirm imaging profiles of the Tm-based probe in NIR-IIb region. The results showed that both the tissue-SBR are similar (~2.2) (Supplementary Fig. 21a). However, the background noise of the MCT camera was lower (Supplementary Fig. 21b) under the same conditions except that the InGaAs camera adopted the high gain mode. Therefore, it suggests that Tm-based probes were excellent NIR-IIb imaging probes, which can also image well on InGaAs cameras. Importantly, it should be noted that the Tm-probe possesses weak emission between 1500–1600 nm, so the detection in the NIR-IIb window using 1500 nm LP mainly starts with 1600 nm. In addition, histological evaluations of mice treated with the Tm probe showed no apparent damage to major organs (Supplementary Fig. 22). Further, a surgically induced middle cerebral artery occlusion (MCAO) model was established to visualize the blood circulatory system dysfunction by employing Tm-probe for NIR-IIc imaging. As a result, the circulatory disorders were successfully visualized, and collateral circulation via a self-protective mechanism was established. (Refer to Supplementary Fig. 23 for details).

Further, to demonstrate the advantage of NIR-IIc imaging detected within 1700–2100 nm, a mixture of Tm(02Er)-NPs@PEG and Er-NPs@PEG probes (of identical concentrations) were injected intravenously into mice and imaged by an MCT camera under 800 nm excitation. As shown in Fig. 4d, e, NIR-IIc imaging had a higher SBR (1.56) than NIR-IIb (SBR = 1.27). Additionally, the tiny blood vessels could be observed with much more clarity in the NIR-IIc window compared to the NIR-IIb (including partial luminescence of Tm³⁺), implying a deeper imaging depth and higher clarity than the Er-based (NIR-IIb) probe. Further, Gastrointestinal (GI) tract imaging was performed after oral gavaging mice a mixture of the two probes of the same luminescence intensity. After 1.5 h of gavage, the NIR-II imaging performance in different sub-windows was evaluated. Compared to the NIR-IIb imaging of the Er-based nanoprobe (SBR = 5.4), a higher SBR (8.6) was observed with the Tm-based probes in the NIR-IIc region (Fig. 4f, g). We believe that fluorescence/luminescence imaging in the 1700–2100 nm sub-window represented a significant breakthrough, as light absorption increased near 1880 nm, photon scattering was minimized and tissue autofluorescence was further suppressed.

Furthermore, in vitro and in vivo imaging were performed to provide a proof-of-concept of the multiwavelength excitation properties using Tm(30Er)-NPs@PEG probe for bioimaging. A 1% intralipid solution was used to verify the NIR-IIc imaging properties of Tm(30Er)-NPs@PEG for multiple excitations. As shown in Supplementary Fig. 24, comparable penetration depths and SBR were observed under the excitation at 800 nm, 980 nm, and 1208 nm. Out of these, the penetration depth was the deepest under the excitation at 800 nm. However, excitation at 1530 nm resulted in the lowest penetration depth (within 3 mm thickness) because of the strong absorption of water. Additionally, GI tract imaging was also carried out. Luminescence signals in NIR-IIc (1750–2250 nm) were recorded 30 min post-gavage. SBR results

(Fig. 4h) revealed that excitation at 800, 980, and 1208 nm afforded excellent NIR-IIc imaging. However, excitation at 1530 nm failed to attain quality wide-field imaging due to the intense absorption peak of water around 1500 nm. Notably, it was recently reported that imaging of inguinal lymph nodes with a penetration depth of ~500 μm was achieved by confocal microscopy in NIR-IIc under 1540 nm or 1650 nm excitation, which showed two-fold deeper penetration than in the 1200–1400 nm range[11]. But, a higher power density of 1540 nm light was needed. These results revealed that extending the excitation and emission to NIR-II could improve the imaging performance if the interference due to water absorption could be overcome by increasing the excitation power or boosting the brightness of the probe. Thus, this multiwavelength excitation and emissions system (Fig. 4i) provided optimal selection for multi-channel imaging/decoupling theranostics.

## Discussion

We have successfully developed an efficient Tm³⁺-based NIR-II luminescence nanoprobe exhibiting emission around 1800 nm, favourable biocompatibility and excellent stability. The nanoprobe underwent dual-wavelength excitation at 800 and 1208 nm and exhibited NIR-IIb to NIR-IIc emission (1600–2100 nm, ³F₄→³H₆). Such unprecedented optical characteristics would provide alternatives for high-contrast deep tissue imaging in vivo. Notably, our results demonstrated the obvious advantage of Tm-probe assisted NIR-IIc imaging over NIR-IIb subwindow in penetration depth and clarity, which could be owing to the restrained scattering background. Importantly, the filters and the emission properties of the imaging probes in this work together determined the actual imaging window. To analyze the advantages of windows in detail, it is necessary to guarantee even emission and detection in the specific window as much as possible in the future research. Still, this work is anticipated that this probe will promote research and development on detectors needed for the NIR-IIc region with higher quantum efficiency.

## Online content

## Methods

### Synthesis of Tm (x% mol) @Y nanoparticles

The Tm³⁺-rich core nanoparticles were typically synthesized by using the following procedure. Taking the LiTmF₄ core as an example, 1 mmol of TmCl₃·6H₂O, 6 mL of OA and 15 mL of ODE were sequentially added to a three-neck flask (100 mL) with stirring under argon gas. After heating to 160 °C and maintaining the temperature until the solid was fully dissolved, the mixture was cooled to room temperature. Next, 104.9 mg of LiOH·H₂O and 148 mg of NH₄F predissolved in 6 mL methanol were added to the above mixture. The solution was heated to 85 °C for 30 min to remove the methanol. After that, the solution temperature was raised to 300 °C and maintained for 1 h. Then, the product was collected by centrifugation at 6500 rpm (4700 g) for 6 min and redispersed in 8 mL of cyclohexane for further use.

The core@shell nanoparticles were prepared by the epitaxial growth method. Furthermore, the shell thickness was tuned with the injection volume of the precursor. Typically, 1 mmol of $Y(OOCCF_3)_3$ and 1 mmol of $LiOOCCF_3$ were placed into a three-neck flask (100 mL), and then 3 mL of OA and 7.5 ml of ODE were added. After complete dissolution of the solid at 100 °C under an argon flow, the solution was cooled to room temperature as a precursor for further use. Afterwards, a specific amount of precursor solution was injected into another flask kept at 290 °C under an argon atmosphere, which contained the $LiTmF_4$ core (preformed batch) dissolved in 3 mL of OA and 7.5 mL of ODE. After that, the reaction was continued for 30 min to obtain Tm-NPs. Subsequent processing steps included cooling, washing, and storage, as mentioned above. Notably, different shell thicknesses of Tm-NPs were obtained by controlling the injection amount.

Similarly, other comparable series of NPs were prepared, including $NaTmF_4@NaYF_4$, $NaYF_4$:20%Yb,1%Tm$@NaYF_4$, $LiErF_4@LiYF_4$, $LiTmF_4$ coating with different shell hosts (Lu, Gd) and Er or Dy doped Tm-NPs were prepared accordingly.

### Surface modification and cytotoxicity assay
PEGylation of Tm-NPs was performed by the thin-film hydration method, which allowed DSPE−PEG$_{2,000}$ to interact with the oleic acid ligand on the surfaces of Tm-based NPs. Typically, 0.125 mmol of Tm(02Er)-NPs dissolved in $CHCl_3$ were transferred to a 10 mL flask containing 50 mg of DSPE-PEG. Afterwards, the mixture was stirred for 30 min at room temperature, and then the $CHCl_3$ was removed by rotary evaporation to form a thin film. A specific volume of deionized water was added to the above flask, and Tm-based probe was formed by hydration at 50 °C for 30 min; this was further purified by filtration through a 0.22 μm filter and centrifugation at 10,000 rpm (7421 g) for 15 min. The as-obtained Tm-based probe was collected for further use. DLS (Zetasizer Software 7.13) was used to monitor the stability of Tm-based NPs. A cytotoxicity assay of the Tm-based probe was conducted by using the standard CCK8 method. Typically, RAW264. 7 and 4T1 cells were seeded in 96-well plates overnight. After confluent growth, different concentrations of probes were placed in wells and incubated for 24 h. Afterwards, CCK8 solution was added to each well, and cell viability (%) was examined with a Bio−Rad microplate reader.

### NIR-IIc imaging in vivo
The Animal Use and Care Committee at Northeastern University approved all of the experimental procedures, which followed the National Institutes of Health Guidelines. BALB/c mice (6–8 weeks old) were provided by Liaoning Changsheng Biotechnology Company for the in vivo study. After intravenously injecting Tm-based NPs, imaging was performed upon 800 nm excitation (~ 100 mW/cm²). The mice were anaesthetized and placed on the imaging stage. In vivo NIR-II imaging was carried out by using the MCT detector (Zephir 2.5, Photon etc) with various filters (longpass filters with a cut-on wavelength of 1500 nm (FELH1,500, Thorlabs), 1600 nm, and 1700 nm were used, respectively) through PHyspecV2 (Photon Etc). A comparison between an InGaAs (SD 640, Tekwin) and MCT camera on background and SBR by vascular imaging of the same sample (Tm(02Er) NPs, 200 μL, ~30 mg/mL) in the NIR-IIb window under 800 nm excitation (200 mW/cm², MCT: 50 ms low gain and InGaAs:50 ms high gain model). In addition, 300 μL of Tm/Er-NPs@PEG was perfused into the mouse's stomach for GI tract imaging. NIR-II imaging signals were collected through a 1700 nm longpass filter combined with a bandpass filter (FB2,000-500, with a transmittance window in 1750–2250 nm). Imaging was recorded with multiwavelength excitation at 800 nm, 980 nm, 1208 nm, and 1530 nm (~120 mW/cm²) with an exposure time of 100 ms. These laser irradiation power below the laser safety limit of NIR lasers, e.g., 800 nm (0.33 W/cm²), 980 nm (0.72 W/cm²), 1208 nm (1 W/cm²) and 1530 nm (1 J/cm²) (American National Standard for Safe Use of Lasers, ANSI Z136.1-2014).

For comparative cerebral vascular imaging and dysfunction of the blood circulatory, ICG dye (200 μL, 100 μg/mL) was intravenously injected into the mouse without MCAO model and then imaged in NIR-IIa subwindow with a 1300 nm LP filter upon 800 nm excitation (~100 mW/cm²). After 2 h ICG metabolism, the MCAO model was established by blocking the right vein with a silicon-coated nylon filament (Cinontech, Beijing) under anaesthesia[34]. After that, the Tm-based NPs (200 μL, 30 mg/mL) were intravenously injected into the mouse, and the brain vessels of the same mouse were imaged in NIR-IIc region under the same excitation conditions with the MCT camera through the 1700 nm LP filter.

### Haematoxylin and eosin (H&E) staining
The major organs were collected and fixed in 4% paraformaldehyde. Afterwards, the organs were further dehydrated, embedded in paraffin, and sectioned into 3 μm thick slides. H&E staining was then performed according to the manufacturer's protocol in the H&E kit, and the stained images were recorded with a microscope (Nikon CS2).

### Statistics and reproducibility
All representative images were performed a minimum of three replicates in independent experiments with similar results.

### Reporting summary
Further information on research design is available in the Nature Portfolio Reporting Summary linked to this article.

## Data availability
The experimental data supporting the findings of this study are available within the article, Supplementary Information, and Source Data. All relative data are available from the corresponding authors upon request. Source data are provided with this paper.

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

## Acknowledgements
This study was financially supported by the National Key Research and Development Program of China (Grant 2021YFA0715603), National Natural Science Foundation of China (Y.C., 62075217, X.K., 11874354, 11874355 and 61575194), Project of Science and Technology Agency, Jilin Province (Y.C., 20210101148JC, 20230508104RC and 202512JC010475440), and the State Key Laboratory of Luminescence and Applications (SKLA-2019-02, SKLA-2020-09). The authors thank Dr. M. Shen, Dr. H. Meng for their help with thrombosis model. The authors acknowledge Dr. B. Yan (Ocean Insight, Shanghai) for assistance with the quantum yield measurements.

## Author contributions
Y.C. conceived and designed the experiments. X. X., H.C., Y.W., W.W., F.W. and R.Y. performed the experiments and H.C., Q.L., X.K., and Y.C. analysed the data and wrote the manuscript. All authors discussed the results and commented on the manuscript.

## Competing interests
The authors declare no competing interests.
