## [Peer Review File · Nature Communications]

Bright Tm3+-based downshifting luminescence nanoprobe operating around 1,800 nm for NIR-IIb and c bioimagingReviewer #1 (Remarks to the Author):

In this work, Chang et al reported a kind of Tm³⁺ doped bright nanocrystal whose emission peaks in the NIR-IIc (1700-1880 nm) window. It is an interesting study with high novelty and significance.

The photoluminescence bioimaging in the second near-infrared imaging window within 900-1700 nm has been verified as a powerful tool for biomedical research. But the imaging probe with emission beyond 1700 nm is very rare in bioimaging. This work provides references to the future advanced bioimaging beyond 1700 nm based on a camera. I recommend this work be published in the journal Nat Comm after drawing the concerns below.

1. The title of this work is "Bright Tm³⁺-based downshifting luminescence nanoprobe operating around 1800 nm for NIR-IIb and c bioimaging", however, limited by the camera used in this work, the imaging applications were still in the NIR-IIb window below 1700 nm. The title is therefore misleading. According to the full text, the keynote was focused on the novel nanocrystal whose emission peaks beyond 1700 nm (there is still a shortage of probes emitting beyond 1600 nm). The claim about the "NIR-IIb and c bioimaging" should not be appeared in the title since the bioimaging involved in this work is below 1700 nm.
2. Line 63-66, Page 4, the author described that "Notably, the detection regions 1700 nm to 1870 nm display absorption and scattering properties similar to those in NIR-IIb; thus, it is defined as the NIR-IIc window and is no longer limited to 1700 nm in the NIR-II window (limited by the classic InGaAs detector)." It would be better to add reference(s) here to make it more convincing.
3. There are still irregularities in writing and typesetting, such as Fig 1b doesn't seem to be mentioned in the main text, and the numbers in Fig. 2f are blocked by Fig. 2g.
4. The emission spectra of LiTmF₄: x%Er@LiYF₄ are shown in Fig. 2f, but the spectrum when x=0 seems not obvious enough.
5. The comparison between the Er and Tm nanoprobe in Fig. 3f&g is confusing. Since the camera cannot respond to the photons beyond 1700 nm, the imaging window of the two nanoprobe is both 1500-1700 nm. So what determines the differences in the images and SBRs? The emission intensities? If the spectral difference between the two nanoprobe affects the imaging performance, it is necessary to add the emission spectra.
6. Are there Er³⁺ doping in the probes used for bioimaging? It was verified that Er³⁺ doping with a doping level below 0.5% could boost the downshifting emission in this work.
7. Authors claimed that "After 5 min of treatment (from 2 to 7 min), the luminescence signals of the right common carotid artery gradually became more potent than those of the left carotid artery (9 min time point), indicating the onset of thrombus formation and increased retention of NPs" (Line 266-269, Page 13). It would be better to add some quantitative results of the signals to make the comparison clearer.
8. Since the intensities of the luminescence signals are affected by the tissue occlusion, concentration (concentrated or diluted) in the body and so on, the description "the intensities of the luminescence signals directly reflect metabolism" (Line 292-294, Page 15) seems not convincing.
9. Control group is needed in the HE test. And the length of the scale bar is missing.
10. The scale bars in Fig. 4g h and j are needed. Supplementary Fig. S18 showed an intensity measurement of ~250 nm. Are the authors sure it's nanometers?
11. The table shown in Fig. 4k needs to be double-checked, such as "optical window".
12. The QE curve of the camera used in this work is also needed since the boundary at ~1700 nm determines which window is used.

Reviewer #2 (Remarks to the Author):

In this manuscript, the authors reported the synthesis of Tm³⁺ doped nanoparticles for

the bioimaging applications in NIR-IIc region. Although the concept is suitable, the contents shown in the manuscript can not fully support the conclusion that NIR-IIc together with the reported probe is a good choice in bioimaging. Thus the current version of manuscript is not suitable to be published. Some main concerns are listed below:

1. The image resolution of the whole text is very low, which affects readers' perception when reading. Some of the images are annotated too small. The graphs corresponding to some data are not beautiful.
2. The authors mentioned the emission intensity would increase by adding Er. However, the manuscript still used pure Tm nanoparticles for subsequent applications. This part of the statement is to explain which nanoparticle's performance is good, and should be used for subsequent application. The authors should choose the best nanoparticle.
3. The concrete luminescence intensity didn't show detail in the whole article. The comparison of the NIR-IIc emission and the other emission also didn't display. So the reader can't know the real relationship of the different emission peaks which is impossible to quantify the advantages of NIR-IIc. The authors need to compare NIR-IIc to other probes directly.
4. The penetration depth of the signal from Tm-NPs was only verified in vitro, but not in vivo. It is not enough to explain the advantages of Tm-NPs in biological imaging. Meanwhile, there was no comparison of the NIR-IIb imaging and NIR-IIc imaging based on Tm-NPs in vivo which didn't show enough investigations of the advantages of the NIR-IIc imaging.
5. The biological model in this paper only establishes the thrombus model to illustrate the imaging effect of the Tm-NPs. A cerebral infarction model would be better here. As for the thrombosis model mentioned in this paper, it only verifies that the formation process does not have clear significance of biomedical research, and is more like simply applying biological model. If the model selection is changed from the formation of thrombus to the treatment of thrombus, it will have more research significance.
6. The entire paper is based on NIR-IIc bioimaging of dominant tissues. The signal-to-noise ratio obtained in vivo imaging is not significantly higher than that of NIR-IIb. At the same time, the image brightness contrast of the two materials in the same particle concentration is not given. So what obvious advantages NIR-IIc imaging has over the existing NIR-IIb imaging is not mentioned in this paper.
7. It is mentioned in the material section and the final biological experiment section that four kinds of excitation light can be used for excitation. If these four excitation lights all have excellent imaging properties, they can be highlighted. However, there is no further comparison of penetration depth and imaging resolution in this paper, which lacks data support. In addition, it is not mentioned that which of the four excitations has the most excellent luminescent properties and is most suitable for application. By the way, many rare earth nanoparticles has the properties of being excited by multiple excitation lights, so it does not make sense to emphasize that it can be excited by four excitation lights without a clear advantage.

Reviewer #3 (Remarks to the Author):

Indeed, fluorescence imaging in the NIR-IIb/c region > 1500 nm can yield high quality images with the potential to overcome the current limitation of special resolution and tissue penetration depth. Unfortunately, there is a relative paucity of NIR-IIb/c probes. To address this shortcoming, the authors report the synthesis of thulium-based rare earth nanoparticles (Tm@Y) which could generate NIR-IIb/c photoluminescence spanning a wide spectral range of 1600-2000 nm. The highly doped Tm³⁺ ion not only increases the emission intensity but also enables multi-wavelength excitation, which could contribute to achieve a number of novel bio-imaging applications. This is indeed a novel NIR-IIb/c luminescent nanoprobe, but the in vivo imaging experiments are insufficient of demonstrating the advantage of this probe in the fluorescence bio-imaging. These imaging were performed in 1500-1700 nm as a 1500 longpass filter and an InGaAs camera with an upper detection limit of 1700 nm were used. Imaging

performance beyond 1700 nm was not demonstrated, but the title, abstract and main text presented imaging in NIR-IIb/c window. There are some deficiencies that need to be corrected as listed below:

1. The authors used an InGaAs detector (640 HS, Princeton Instruments) in this work. The upper spectral detection limit of InGaAs detectors is ~ 1700 nm, which was also mentioned in the manuscript and confined the fluorescence imaging at < 1700 nm window. This is a contradiction to the title "operating around 1800 nm for NIR-IIb and c bioimaging". If the authors want to keep the title as it is, they have to do bioimaging around 1800 nm. The definition is also confusing, NIR-IIb is 1500-1700 nm
2. There is a recent work by the Dai group in Nature Nanotechnology on NIR-IIc imaging using QDs and single photon detectors. The authors should mention it and compare with data in that work. The authors should make the various subwindows of NIR-II to be consistent with the ones in that work.
3. The quantum yield of the Tm@Y was measured by using ICG dye as a reference standard, which was improper due to the huge difference between their emission wavelengths (~ 1000 nm). The absolute QY of the Tm@Y should be measured.
4. The authors synthesized both the Li-based LiTmF₄@LiYF₄ NPs and the Na-based NaTmF₄@NaYF₄ NPs, which one is brighter? Also, is the LiTmF₄@LiYF₄ NPs brighter than the traditional NaYF₄:20%Yb,1%Tm@NaYF₄ NPs in the NIR-IIb/c region?
5. In Supplementary Fig. 2a, the dried Tm-NPs still has an emission peak at ~ 1680 nm. It is a contradiction to the statement of "a single emission band at approximately 1680 nm was not observed from powdered Tm³⁺-doped NPs".
6. In Fig. 1f, I did not see patterns in the "Y" image as shown in "Tm" image. How does the image of "F" look like?
7. Which InGaAs detector was used to measure the emission spectra shown in Supplementary Fig. 3? Why is there no signal after 1700 nm?
8. There is no black curve corresponding to "x=0" in Fig. 2f.
9. The Fig. 2f-i demonstrated the luminescence properties of LiTmF₄:x%Er@LiYF₄ NPs (LiYF₄ shell), but the Supplementary Fig. 8 showed the TEM images of LiTmF₄:x%Er@LiLuF₄ NPs (LiLuY₄ shell). Why?
10. The lifetime of Tm@Y is very confusing. In Fig. 2b, the lifetime of LiTmF₄@LiYF₄ (when x% Tm = 100) is 19.1 ms. But in Fig. 2g, the lifetime of LiTmF₄@LiYF₄ (when x% Er = 0) is 2.5 ms. Which one is real?
11. The authors claimed "Interestingly, the luminescence intensity of Tm-NPs@PEG in aqueous solution decreased much more than it did in cyclohexane (Fig. 3e)". In which window? It is confusing
12. In Fig. 3e, how much the luminescence intensity of the Tm@Y in aqueous solution decreased compared with in cyclohexane?
13. In Fig. 3, the authors compared the in vitro imaging in the NIR-IIb (Er-NPs) and the NIR-IIc (Tm-NPs) windows. The authors should also compare the in vivo imaging in the NIR-IIb and the NIR-IIc windows in Fig. 4, by using these two probes.
14. 1600 nm or 1700 nm LP filter should be used instead of 1500 nm LP filter in Fig. 4 to ensure the NIR images taken in the NIR-IIb/c region of 1600-2100 nm.
15. The authors claimed "A bright, clear vascular arrangement appeared in the mouse body (Fig. 4a)." I did not see vascular in Fig. 4a. Was the skin removed? If so, the authors need to give more details about this experiment.
16. The definition of optical window of NIR-I and NIR-II in Fig. 4k is confusing.
17. How is the Tm-NPs@PEG circulation in mouse body after intravenous injection? It is important to know for in vivo imaging.
18. In Supplementary Fig. 5, histograms demonstrating the particle size distribution are needed.
19. Line 149, the "linearly" is not correct.
20. The reference 9 and 14 are the same.
21. Check the format of reference 3, 5, 13, 25 and 30.

Responses to the reviewers' comments (Manuscript number: NCOMMS-22-15435-T for Nature Communications)

Title: Bright Tm³⁺-based downshifting luminescence nanoprobe operating around 1800 nm for NIR-IIb and c bioimaging

For editor and reviewers' convenience, we have listed below the new experiments and major advances made in the revision:

As suggested by reviewer #1, To fully reflect the imaging performance of Tm-based nanoprobe beyond 1700 nm, we have supplemented bioimaging experiments using an HgCdTe (MCT) camera whose response wavelength is from 850 nm to 2.5 μm , replacing the results collected by InGaAs camera. Experimental results indicate that NIR-IIc imaging showed better performances than NIR-IIb.

As suggested by reviewer #2, we designed new experiments to supplement the in vitro and in vivo NIR-IIc imaging using a combination of an MCT camera and various filters. The results demonstrated a higher signal-to-background ratio and higher resolution of NIR-IIc imaging, and Tm-based NPs would serve as excellent NIR-IIc candidates.

As suggested by reviewer #3, In addition to supplementing in vivo imaging experiments in the NIR-IIc region to demonstrate the advantage of the Tm-based probe, we also supplemented some characterization results for Tm-NPs.

Our point-by-point responses (in blue) to the reviewers' comments (in black) are provided below.

Reviewer #1: In this work, Chang et al reported a kind of Tm³⁺ doped bright nanocrystal whose emission peaks in the NIR-IIc (1700-1880 nm) window. It is an interesting study with high novelty and significance.

The photoluminescence bioimaging in the second near-infrared imaging window within 900-1700 nm has been verified as a powerful tool for biomedical research. But the imaging probe with emission beyond 1700 nm is very rare in bioimaging. This work provides references to the future advanced bioimaging beyond 1700 nm based on a camera. I recommend this work be published in the journal Nat Comm after drawing the concerns below.

Response: We thank the reviewer for the constructive comments and the appreciation of our work. We address the reviewer's comments and questions in the following.

1. The title of this work is "Bright Tm³⁺-based downshifting luminescence nanoprobe operating around 1800 nm for NIR-IIb and c bioimaging", however, limited by the camera used in this work, the imaging applications were still in the NIR-IIb window below 1700 nm. The title is therefore misleading. According to the full text, the keynote was focused on the novel nanocrystal whose emission peaks beyond 1700 nm (there is still a shortage of probes emitting beyond 1600 nm). The claim about the "NIR-IIb and c bioimaging" should not be appeared in the title since the bioimaging involved in this work is below 1700 nm.

Response: Thanks for your suggestion. In fact, the NIR camera (InGaAs) we originally used is responsive beyond 1700 but has lower quantum efficiency. To fully reflect the imaging performance of Tm-based nanoprobe beyond 1700 nm, we have supplemented bioimaging experiments using an HgCdTe (MCT) camera whose response wavelength is from 850 nm to 2.5 μm. We believe the current results are consistent with the title and not misleading.

2. Line 63-66, Page 4, the author described that "Notably, the detection regions 1700 nm to 1870 nm display absorption and scattering properties similar to those in NIR-IIb;

thus, it is defined as the NIR-IIc window and is no longer limited to 1700 nm in the NIR-II window (limited by the classic InGaAs detector)." It would be better to add reference(s) here to make it more convincing.

Response: Thanks for the suggestion. Two relevant references for these claims were cited in the revised manuscript.

Ref. 1: Feng, Z., Tang, T., Wu, T. et al. Perfecting and extending the near-infrared imaging window. *Light Sci Appl* 10, 197 (2021). [https://doi.org/ 10.1038/s41377-021-00628-0](https://doi.org/10.1038/s41377-021-00628-0).

Ref. 2: Wang, F., Ren, F., Ma, Z. et al. In vivo non-invasive confocal fluorescence imaging beyond 1,700 nm using superconducting nanowire single-photon detectors. *Nature Nanotech.*,

3. There are still irregularities in writing and typesetting, such as Fig 1b doesn't seem to be mentioned in the main text, and the numbers in Fig. 2f are blocked by Fig. 2g.

Response: We have changed the writing and typography irregularities in the revised manuscript.

4. The emission spectra of $\text{LiTmF}_4: x\% \text{Er} @ \text{LiYF}_4$ are shown in Fig. 2f, but the spectrum when $x=0$ seems not obvious enough.

Response: Thanks for your reminder. The emission intensity of $\text{LiTmF}_4: 1\% \text{Er} @ \text{LiYF}_4$ is close to $\text{LiTmF}_4 @ \text{LiYF}_4$, causing the emission spectrum of $\text{LiTmF}_4 @ \text{LiYF}_4$ (in black) is not obvious enough.

Indeed, these emission spectra are from $\text{LiTmF}_4: x\% \text{Er} @ \text{LiLuF}_4$ instead of $\text{LiTmF}_4: x\% \text{Er} @ \text{LiYF}_4$, which can be confirmed by the corresponding TEM images for clues as shown in the supplementary Figure S10. Although the LiLuF_4 shell can also reveal the changing law of luminescence with Er^{3+} doping, as well as the comparable brightness to the LiYF_4 shell, to maintain the correlation with the previous spectra results of varying Tm^{3+} concentrations ($\text{LiYF}_4: x\% \text{Tm} @ \text{LiYF}_4$, as shown in Fig 2a), we provided a new relevant data of $\text{LiTmF}_4: x\% \text{Er} @ \text{LiYF}_4$ in the revised manuscript Fig 2f, and all the spectra are also distinguishable.

Original	Revision
-----------------	-----------------

5. The Comparison between the Er and Tm nanoprobes in Fig. 3f&g is confusing. Since the camera cannot respond to the photons beyond 1700 nm, the imaging window of the two nanoprobes is both 1500-1700 nm. So what determines the differences in the images and SBRs? The emission intensities? If the spectral difference between the two nanoprobes affects the imaging performance, it is necessary to add the emission spectra.

Response: Thanks for your careful reading, and we apologize for any confusion. We believe that the spectral difference affects the imaging performance.

As we all know, the spectral response bands of the InGaAs camera can be adjusted by varying the doping composition. For example, this InGaAs detector can respond to the emission of Tm NPs around 1700 nm but with low QE. In addition, although the imaging window of the two nanoprobes is both 1500-1700 nm, which does not represent within the same window that will have the same imaging depth and clarity. For example, the water absorption coefficient around 1500 nm is larger than that of ~1700 nm, and there are also over 100 nm Stokes shifts spanning from the emission of Er^{3+} (1525 nm) to Tm^{3+} (1680 nm), as well as the light scattering is further reduced.

Furthermore, we re-performed these experiments using the HgCdTe (MCT) camera in the revised manuscript and provided the corresponding results in Fig. 3f&g accordingly, including their emission spectra (Fig. 3e). We compared the penetration depth by adjusting the two probes to the same intensity through the camera. These results suggested that the difference between the Er- and Tm-based nanoprobes is caused by the emission bands of the two nanoparticles.

Figure 3e. Emission spectra of Tm(0.2Er)-NPs@PEG and Er-NPs@PEG aqueous solution with the same intensity under the MCT camera upon 800 nm excitation.

6. Are there Er^{3+} doping in the probes used for bioimaging? It was verified that Er^{3+} doping with a doping level below 0.5% could boost the downshifting emission in this work.

Response: The probes used for biological imaging in the original manuscript were Tm NPs with 0.2% Er^{3+} doping, except that Figure 4j is a 30% Er^{3+} doped nanoprobe that can respond to multiple excitations.

To demonstrate better NIR-IIc imaging performance, we performed *in vivo* and *in vitro* imaging experiments using Tm(0.2Er)@PEG nanoprobe coupled with the HgCdTe (MCT) detector. The experimental results were supplemented in the revised manuscript and discussed accordingly.

Original: InGaAs camera	Revision: MCT camera
-----------------------------

7. Authors claimed that "After 5 min of treatment (from 2 to 7 min), the luminescence signals of the right common carotid artery gradually became more potent than those of the left carotid artery (9 min time point), indicating the onset of thrombus formation and increased retention of NPs" (Line 266-269, Page 13). It would be better to add some quantitative results of the signals to make the comparison clearer.

Response: Thanks for your suggestion. To remove confusion caused by the results with the InGaAs camera, we re-performed the relevant experiments using an MCT camera whose spectral response range (850-2500 nm) based on the reviewers' comments. In addition, the quantification of the relevant luminescence signals is provided in the revised manuscript, making the comparison clearer.

Supplementary Fig. S21. NIR-II imaging was obtained using an MCT camera in healthy mice with ICG for brain vascular imaging as the control in the NIR-IIa subwindow (1300 LP filter), and NIR-IIc imaging of cerebral infarction models with Tm-based NPs with a 1700 LP filter at different time points under 800 nm excitation. Scale bars are 10 mm.

8. Since the intensities of the luminescence signals are affected by the tissue occlusion, concentration (concentrated or diluted) in the body and so on, the description "the intensities of the luminescence signals directly reflect metabolism" (Line 292-294, Page 15) seems not convincing.

Response: We thank the reviewer for the valuable comments. We agree with the reviewer. The expression that the intensity of the luminescent signal directly reflects the metabolism is indeed not rigorous enough, which should be corrected.

However, as suggested by the reviewer's comments, we redesigned this experiment to reveal better the NIR-IIc imaging performance than NIR-IIb using the MCT camera (spectral response curve from 850 – 2500 nm), and the Er and Tm probes with the same intensity were used. Thus, this part of the discussion has been removed from the revised manuscript. The corresponding results and discussion were provided in the in vivo section, as shown in Fig.4f

9. Control group is needed in the HE test. And the length of the scale bar is missing.

Response: The control group in the HE test was provided, as shown in the supplementary information in Figure 19, and we added the scale bars accordingly.

Supplementary Fig. S20. H&E staining of major organs after treatment with PBS and Tm(0.2Er) NPs@PEG nanoprobe.

10. The scale bars in Fig. 4g h and j are needed. Supplementary Fig. S18 showed an intensity measurement of ~250 nm. Are the authors sure it's nanometers?

Response: We thank the reviewer for pointing this out. We have added the scale bar to the images and corrected the error. However, the new MCT camera was employed for in vivo imaging, the original data was updated, and corresponding scale bars and fitting

information were provided accordingly.

11. The table shown in Fig. 4k needs to be double-checked, such as "optical window."

Response: We have corrected the error in Fig 4k and checked others throughout the manuscript.

12. The QE curve of the camera used in this work is also needed since the boundary at ~1700 nm determines which window is used.

Response: The QE curve of the camera is provided below. Although the QE gradually decreases beyond 1700 nm, luminescence signals can still be detected. In addition, the new MCT detector was used in the revised manuscript.

Reviewer #2:

In this manuscript, the authors reported the synthesis of Tm³⁺ doped nanoparticles for the bioimaging applications in NIR-IIc region. Although the concept is suitable, the contents shown in the manuscript can not fully support the conclusion that NIR-IIc together with the reported probe is a good choice in bioimaging. Thus the current version of manuscript is not suitable to be published. Some main concerns are listed below:

Response: We thank the reviewers for their affirmation of the concepts in our manuscript. We took the valuable comments very seriously and have tried to address them in the following.

1. The image resolution of the whole text is very low, which affects readers' perception when reading. Some of the images are annotated too small. The graphs corresponding

to some data are not beautiful.

Response: We adjusted the resolution and the size of the annotation font of the images. In addition, Fig. 1 and Fig. 3 were rearranged for beauty and readability, as well as the data in Fig. 4 have been updated and adjusted accordingly. We think they are better now.

2. The authors mentioned the emission intensity would increase by adding Er. However, the manuscript still used pure Tm nanoparticles for subsequent applications. This part of the statement is to explain which nanoparticle's performance is good, and should be used for subsequent application. The authors should choose the best nanoparticle.

Response: Thanks for the suggestion. In fact, the Tm NPs we used are the mentioned optimized ones (0.2% Er³⁺ doped Tm-NPs). In order to simplify the comparison with Er-NPs, ensure smoothness and avoid confusion throughout the article in describing the novel Tm-NPs, we did not emphasize the doping of 0.2% Er³⁺ in bioimaging but also to distinguish it from the four-wavelength excitation of LiTmF₄:30% Er@LiYF₄ NPs (Tm/Er-NPs). In the revised manuscript, we have further clarified the abbreviations, e.g., Tm:0.2%Er@Y NPs short as Tm(02Er) NPs and LiTmF₄:30% Er@LiYF₄ short as Tm(30Er)-NPs and re-updated the relevant data.

It is important to note that we mainly proposed a new type of Tm-based luminescent nanoprobe for imaging in the NIR-IIb/c subwindow, and low-level doping (below 1% Er³⁺) could boost the Tm³⁺ emission. Therefore, following your suggestion, we supplemented the *in vivo* imaging results of low Er³⁺ (0.2%) doped Tm-NPs for NIR-IIc imaging under 800 nm light excitation. Further, 30% Er³⁺ doped Tm-NPs probe was employed under four wavelength excitations using an MCT camera for NIR-IIc imaging.

3. The concrete luminescence intensity didn't show detail in the whole article. The comparison of the NIR-IIc emission and the other emission also didn't display. So the reader can't know the real relationship of the different emission peaks which is impossible to quantify the advantages of NIR-IIc. The authors need to compare NIR-IIc to other probes directly.

Response: Thanks for your valuable suggestion. Since there are few reports of luminescence probes in the NIR-IIc region, we chose the commonly used Er-based NPs

whose peak emission is in the NIR-IIb region (1525 nm). In particular, highly doped single Er^{3+} core/shell NPs are a recently developed high-efficiency NIR-IIb luminescent probe; thus, $\text{LiErF}_4@ \text{LiYF}_4$ NPs with the same host (LiLnF_4) and shell thickness were employed for a relatively fair comparison. The emission spectra of Tm- and Er-NPs PEGylated aqueous solutions with the same concentration were collected by an InAs detector upon 800 nm excitation, as shown in Figure 3e.

Notably, Tm-based nanoprobe can span NIR-IIb to NIR-IIc, ensuring comparability with Er-NPs (1525 nm emission, NIR-IIb). Furthermore, to visually compare the luminescence imaging profiles of Er-NPs and Tm-based NPs in different subwindows, we imaged the as-mentioned samples with the same concentration using the MCT camera under 800 nm excitation, as shown in Fig. 3f. The above results indicate that Tm-based NPs is a bright NIR-IIc luminescence probe for bioimaging.

Figure 3e. Emission spectra of Tm(02)-NPs@PEG and Er-NPs@PEG aqueous solution with the same intensity under the MCT camera upon 800 nm excitation.

Figure 3f. Tm(02Er)-NPs@PEG and Er-NPs@PEG nanoparticles in vitro under 800 nm excitation with different filters (corresponding transmission profiles were provided in Supplementary Fig.17).

4. The penetration depth of the signal from Tm-NPs was only verified in vitro, but not in vivo. It is not enough to explain the advantages of Tm-NPs in biological imaging.

Meanwhile, there was no comparison of the NIR-IIb imaging and NIR-IIc imaging based on Tm-NPs *in vivo* which didn't show enough investigations of the advantages of the NIR-IIc imaging.

Response: Thanks for the valuable comments. In the revised manuscript, we supplemented the *in vitro* and *in vivo* imaging experiments of Tm and Er-based nanoprobe using the MCT camera to investigate the advantages of NIR-IIc imaging.

To verify the penetration depth of the signal from Tm³⁺ emission *in vitro*, the same downshifting luminescence intensity of Tm and Er-based nanoprobe under 800 nm excitation were separately filled in the capillaries and immersed in 1% lipid solution at different depths for imaging performance comparison. As shown in Fig. 3g and h, compared to NIR-IIb imaging (Er-NPs), Tm-based NPs show narrower FWHM and higher SBR results in NIR-IIc, indicating higher imaging clarity of NIR-IIc imaging.

Furthermore, we mixed Tm- and Er-based probes at the same concentration and injected them into the mice via the tail vein. After that, the NIR-II imaging of mice was observed by an MCT camera under 800 nm excitation. As shown in Fig. 4d and e, NIR-IIc imaging (1700 nm LP) has a higher resolution (SBR=1.56) than NIR-IIb imaging (1500 nm LP + 1700 SP, SBR=1.27). Notably, compared to NIR-IIb imaging (although including Tm-NPs luminescence in NIR-IIb), the signals from tiny blood vessels can be more clearly observed in the NIR-IIc window. These results demonstrated that the Tm-based (NIR-IIc) nanoprobe has a deeper imaging depth and higher clarity than the Er-based one (NIR-IIb).

Fig. 3. Luminescence images of a capillary tube filled with Tm(0.2Er)-NPs@PEG and

Er-NPs@PEG solution, immersed in various thicknesses of 1% intralipid solution recorded Er³⁺ or Tm³⁺ emissions beyond 1500 nm upon 800 nm excitation, respectively. h, Corresponding full width at half maximum (FWHM) and SBR luminescence intensity profiles.

Fig.4d, Vascular luminescence imaging with Tm- and Er-based NPs at the same concentration in NIR-IIb (1,500 nm LP+1,700 nm SP filters,1500-1700 nm) and NIR-IIc (1,700 nm LP filter) subwindow. **e**, corresponding luminescence intensity statistics of **d**.

5. The biological model in this paper only establishes the thrombus model to illustrate the imaging effect of the Tm-NPs. A cerebral infarction model would be better here. As for the thrombosis model mentioned in this paper, it only verifies that the formation process does not have clear significance of biomedical research, and is more like simply applying biological model. If the model selection is changed from the formation of thrombus to the treatment of thrombus, it will have more research significance.

Response: Thanks for the suggestion. As you suggested, we have performed a

cerebral infarction model to illustrate the imaging effect of the Tm(02Er) NPs using an MCT camera. In addition, compared to the effective thrombus treatment period, the blood pool time of nanoprobe is shorter, making it difficult to obtain valuable imaging results. Thus, we only did the cerebral infarction model experiment, and the corresponding results are provided in the revised manuscript and discussed accordingly, as shown in **Supplementary Fig. 23**.

Supplementary Fig. S23. NIR-II imaging was obtained using an MCT camera in healthy mice with ICG for brain vascular imaging as the control in the NIR-IIa subwindow (1300 LP filter), and NIR-IIc imaging of cerebral infarction models with Tm-based NPs with a 1700 LP filter at different time points under 800 nm excitation. Scale bars are 10 mm.

6. The entire paper is based on NIR-IIc bioimaging of dominant tissues. The signal-to-noise ratio obtained in vivo imaging is not significantly higher than that of NIR-IIb. At the same time, the image brightness contrast of the two materials in the same particle concentration is not given. So what obvious advantages NIR-IIc imaging has over the existing NIR-IIb imaging is not mentioned in this paper.

Response: Thanks for your valuable comments. Following your suggestion, we performed *in vivo* and *in vitro* imaging experiments using an MCT camera under 800 nm excitation. As shown in Fig. 3f, Er and Tm nanoprobe with the same concentration are imaged in different subwindows matched with various filters. Er- and Tm-based probes can be distinguished separately by 1500 BP (1524-1536 nm) and 1700 LP filters.

The combination of 1500 LP with 1700 SP filters (1500-1700 nm) can clearly see the luminescence brightness of these two imaging probes in the NIR-IIb region, indicating that Tm-based NPs can be used as NIR-IIb probes. With the longpass filters changing from 1500, 1600, to 1700 nm, the Er-NPs gradually become undetectable, but the bright Tm-based NPs remain, confirming the NIR-IIc imaging profiles of the Tm-based probe. Next, we performed the bioimaging experiment to show the advantage of NIR-IIc imaging under 800 nm excitation using the MCT camera, as shown in Fig. 4d and e. The vascular imaging results show that NIR-IIc imaging (SBR=1.56) has higher SBR and resolution than NIR-IIb (SBR=1.27), indicating obvious advantages of NIR-IIc imaging.

Figure 3f. Tm(02Er)-NPs@PEG and Er-NPs@PEG nanoparticles in vitro under 800 nm excitation with different filters (corresponding transmission profiles were provided in Supplementary Fig.18).

Fig.4d. Vascular luminescence imaging with Tm- and Er-based NPs at the same concentration in NIR-IIb (1,500 nm LP+1,700 nm SP filters,1500-1700 nm) and NIR-IIc (1,700 nm LP filter) subwindow. **e**, corresponding luminescence intensity statistics of **d**.

7. It is mentioned in the material section and the final biological experiment section that four kinds of excitation light can be used for excitation. If these four excitation lights all have excellent imaging properties, they can be highlighted. However, there is no further comparison of penetration depth and imaging resolution in this paper, which lacks data support. In addition, it is not mentioned that which of the four excitations has the most excellent luminescent properties and is most suitable for application. By the way, many rare earth nanoparticles has the properties of being excited by multiple excitation lights, so it does not make sense to emphasize that it can be excited by four excitation lights without a clear advantage.

Response: Thanks for your valuable suggestions. We supplemented the NIR-IIc imaging using Tm(30Er)-NPs under four-wavelength excitations.

In vitro and *in vivo* imaging were performed to provide a proof-of-concept of the multiwavelength excitation properties using Tm(30Er)-NPs@PEG probe for bioimaging. A 1% intralipid solution was used to verify the NIR-IIc imaging properties of Tm(30Er)-NPs@PEG for multiple excitations. As shown in Supplementary Fig. 22, similar penetration depths and SBR were observed under the excitation at 800 nm, 980 nm, and 1208 nm. Out of these, the penetration depth was the deepest under the excitation at 800 nm. However, excitation at 1530 nm resulted in the lowest penetration depth (within 3 mm thickness) because of the strong absorption of water. In addition, we also demonstrated the intragastric imaging of mice under four excitation lights, including NIR-II excitation (1208 and 1530 nm) and NIR-IIc emission realized in this system. Fig. 4 h showed that luminescence signals beyond 1700 nm (1,750-2,250 nm) were observed for the stomach and the intestines at 30 min postgavage with the same excitation powers (120 mW/cm^2) at 800 nm, 980 nm, 1,208 nm, and 1,530 nm, respectively. SBR results (Fig. 4i) showed that these three excitation lights (800, 980, and 1,208 nm) afforded excellent imaging properties in NIR-IIc, but due to the intense absorption peak of water around 1,500 nm, 1,530 nm excitation fails to show a clearer imaging quality. Nevertheless, this multiwavelength excitation and emissions system provided optimal selection, which would play a remarkable role in multi-channel imaging/decoupling theranostics for precise and safe treatment.

Supplementary Figure S24. Luminescence images of a capillary tube filled with Tm(30Er) NPs solution, immersed in 1% intralipid solution at different depths (1 mm, 2 mm, 3 mm, 4 mm, 5 mm and 6 mm) and collected at NIR-IIc region (1750-2250 nm)

upon 800 nm, 980 nm, 1,208 nm and 1,530 nm excitation, respectively. And the corresponding SBR luminescence intensity profiles.

Fig.4 h, Gastrointestinal (GI) tract imaging in NIR-IIc (1,750-2250 nm) region in living mice gavaged with Tm(30Er)-NPs@PEG upon 800 nm, 980 nm, 1,208 nm, and 1,530 nm irradiation (120 mW/cm²), respectively. **i**, corresponding SBR statistics of **h**.

Reviewer #3: Indeed, fluorescence imaging in the NIR-IIb/c region > 1500 nm can yield high quality images with the potential to overcome the current limitation of special resolution and tissue penetration depth. Unfortunately, there is a relative paucity of NIR-IIb/c probes. To address this shortcoming, the authors report the synthesis of thulium-based rare earth nanoparticles (Tm@Y) which could generate NIR-IIb/c photoluminescence spanning a wide spectral range of 1600-2000 nm. The highly doped Tm³⁺ ion not only increases the emission intensity but also enables multiwavelength excitation, which could contribute to achieve a number of novel bio-imaging applications. This is indeed a novel NIR-IIb/c luminescent nanoprobe, but the in vivo imaging experiments are insufficient of demonstrating the advantage of this probe in the fluorescence bio-imaging. These imaging were performed in 1500-1700 nm as a

1500 longpass filter and an InGaAs camera with an upper detection limit of 1700 nm were used. Imaging performance beyond 1700 nm was not demonstrated, but the title, abstract and main text presented imaging in NIR-IIb/c window. There are some deficiencies that need to be corrected as listed below:

Response: We appreciate these positive views on our manuscript and the comments, which have helped up improve the paper significantly. We have performed additional experiments to investigate imaging performance beyond 1700 nm using an MCT camera whose responsive wavelength spans 850 to 2.5 μ m nm and revised our manuscript accordingly.

1. The authors used an InGaAs detector (640 HS, Princeton Instruments) in this work. The upper spectral detection limit of InGaAs detectors is \sim 1700 nm, which was also mentioned in the manuscript and confined the fluorescence imaging at $<$ 1700 nm window. This is a contradiction to the title "operating around 1800 nm for NIR-IIb and c bioimaging". If the authors want to keep the title as it is, they have to do bioimaging around 1800 nm. The definition is also confusing, NIR-IIb is 1500-1700 nm

Response: Thanks for the insightful comments and suggestions. We have re-performed the in vivo bioimaging using Tm (02Er)- NPs and MCT camera operating around 1800 nm using 1700 LP (Customized) and 2000-500 nm BP filter (1750-2250 nm, Thorlabs).

In addition, confusion about the definition of NIR-IIb subwindows (1500-1700 nm), which is mainly influenced by the upper spectral detection limit of conventional InGaAs detectors, and several research groups have carried out analogous definitions. For example, Dai, Qian and Tang et al. defined the long wavelength range of NIR-II (1500 nm-1700 nm) as the NIR-IIb subwindow. The refs are as below.

1) Ref.1: Diao, S., Hong, G., Antaris, A.L. et al. Biological imaging without autofluorescence in the second near-infrared region. *Nano Res.* 8, 3027–3034 (2015). <https://doi.org/10.1007/s12274-015-0808-9>;

2) Ref. 2: Li, Y., Cai, Z., Liu, S. et al. Design of AIEgens for near-infrared IIb imaging through structural modulation at molecular and morphological levels. *Nat Commun* 11,

1255 (2020). <https://doi.org/10.1038/s41467-020-15095-1>;

3) Ref.3: Alifu, N., Zebibula, A., Zhang, H. et al. NIR-IIb excitable bright polymer dots with deep-red emission for in vivo through-skull three-photon fluorescence bioimaging. *Nano Res.*13, 2632–2640 (2020). <https://doi.org/10.1007/s12274-020-2902-x>.

2. There is a recent work by the Dai group in *Nature Nanotechnology* on NIR-IIc imaging using QDs and single photon detectors. The authors should mention it and compare with data in that work. The authors should make the various subwindows of NIR-II to be consistent with the ones in that work.

Response: We note that Dai group in *Nature Nanotechnology* gave a detailed definition of imaging subwindows of NIR-II. Although the wide-field imaging we performed was different from the confocal microscopy imaging, and the different material systems were also used (QDs and RENCs), we all demonstrated that the imaging depth and SBR of NIR-IIc are better than that of NIR-IIb region (see imaging section for details). Therefore, we have made the suggested change, including citing it.

3. The quantum yield of the Tm@Y was measured by using ICG dye as a reference standard, which was improper due to the huge difference between their emission wavelengths (~1000 nm). The absolute QY of the Tm@Y should be measured.

Response: Indeed, we also wanted to test absolute quantum effects before. However, the response range of currently available integrating spheres is 230-1700 nm. Therefore, the measurement error is also huge, and there is no relevant calibration curve at present.

Therefore, in this paper, we measured the relative quantum efficiency and used two reference dyes as references [Refs. a and b.]. Due to the low quantum efficiency of IR-26 (0.05%-0.5%), especially the weak luminescence under 800 nm excitation, resulting in a relatively larger standard deviation. Thus, we chose the ICG dye (12%) as the reference dye. The relative QY values of several key samples are shown in the table.

Samples	Size of core (nm)	Size of core/shell (nm)	Relative QY based on ICG	Relative QY based on IR-26
---------	----------------------	----------------------------	-----------------------------	-------------------------------

LiTmF ₄ @LiYF ₄	~ 11.8×15.8	~ 18×30	~ 32.9 ± 13.2%	~ 44.8 ± 21.5%
LiTmF ₄ @LiYF ₄	~ 9.8×13.7	~ 19.3×24.8	~ 20.6 ± 4.1%	~ 23.3 ± 13.3%
LiErF ₄ @LiYF ₄	~ 10.9×14.4	~ 19.95× 24.76	~ 25.3 ± 4.6%	~ 33.2 ± 20.8%
LiTmF ₄ :02%Er @LiYF ₄	~ 10.4×14.4	~ 18.88× 23.42	~ 33.5 ± 9.5%	~ 54.2 ± 20.8%

* QY-ICG=12%^a, QY-IR26=0.05%^b

Ref. a. Benson RC, Kues HA. Fluorescence properties of indocyanine green as related to angiography. *Physics in Medicine and Biology* 1978, 23(1): 159-163.

Ref. b. Absolute Photoluminescence Quantum Yields of IR-26 Dye, PbS, and PbSe Quantum Dots. DOI: 10.1021/jz100830r | *J. Phys. Chem. Lett.* 2010, 1, 2445–2450.

4. The authors synthesized both the Li-based LiTmF₄@LiYF₄ NPs and the Na-based NaTmF₄@NaYF₄ NPs, which one is brighter? Also, is the LiTmF₄@LiYF₄ NPs brighter than the traditional NaYF₄:20%Yb,1%Tm@NaYF₄ NPs in the NIR-IIb/c region?

Response: The emission spectra of Li-based LiTmF₄@LiYF₄ NPs, Na-based NaTmF₄@NaYF₄ NPs and traditional NaYF₄:20%Yb,1%Tm@NaYF₄ NPs with the same concentration are shown in Supplementary Fig. 17e. The downshifting luminescence of Tm-NPs originates from its intense cross-relaxation process. Due to the severe energy back transfer process from Tm to Yb, and the low concentration of Tm-doped in NaYF₄:20%Yb,1%Tm@NaYF₄ NPs (TEM as shown in the following Supplementary Figure 17d), the NIR-IIb/c emission of NaYF₄:20%Yb,1%Tm@NaYF₄ NPs is relatively weak with same power density but different excitation lights (800 nm for Tm@Y NPs and 980 nm for Yb/Tm NPs).

In addition, the downshifting luminescence of Li-based NPs is stronger than that of Na-based NPs may be due to the intense crystal field polarization of Li host materials.

Supplementary Fig. 17d, TEM images of NaYF₄:20%Yb,1%Tm@NaYF₄ core and NaYF₄:20%Yb,1%Tm@NaYF₄ core-shell NPs with the size of 26.02 ± 2.1 nm. e, Comparison of emission profiles of Tm-based NPs with a similar size using the same excitation power density (980 nm light for Yb/Tm NPs, 800 nm light for Na- and Li-based Tm-NPs).

5. In Supplementary Fig. 2a, the dried Tm-NPs still has an emission peak at ~ 1680 nm. It is a contradiction to the statement of "a single emission band at approximately 1680 nm was not observed from powdered Tm³⁺-doped NPs".

Response: Thanks for your suggestion. We have made the change in this revised manuscript. Cyclohexane as a solvent severely quenches the luminescence around 1,680–1,900 nm than other bands due to strong NIR-IIc absorption of cyclohexane (Supplementary Fig. 3b).

6. In Fig. 1f, I did not see patterns in the "Y" image as shown in "Tm" image. How does the image of "F" look like?

Response: Thanks for your careful reading and reminder. We re-rested the TEM-related characterization. We believe that the mapping patterns of "Y" is better now and supplemented the image of "F", as shown in Fig 4. In addition, as suggested by the reviewers, we rearranged Fig.1 to make it more readable and beautiful.

7. Which InGaAs detector was used to measure the emission spectra shown in Supplementary Fig. 3? Why is there no signal after 1700 nm?

Response: This InGaAs detector is InGaAs Stesta, Model No-NIR300/2. This manuscript contained three detectors for spectral measurement, including red PMT for 190-800 nm, InGaAs for 1000-1700 nm, and InAs for \sim 1000-3300 nm, respectively. To avoid confusion for the reader, we provided the new results from the InAs detector, and deprecated InGaAs measured results.

8. There is no black curve corresponding to "x=0" in Fig. 2f.

Response: The black (x=0) and red (x=1) lines are nearly coincident, so they cannot be clearly distinguished. And please also see our response to Reviewer 1, Comments 4.

"Indeed, these emission spectra are from $\text{LiTmF}_4: x\% \text{Er} @ \text{LiLuF}_4$ instead of

LiTmF₄: x%Er@LiYF₄, which can be seen from the corresponding TEM images for clues as shown in the supplementary information of Figure 2f. Although the LiLuF₄ shell can also reveal the law after doping with Er³⁺, to maintain the correlation with the previous spectra results of varying Tm³⁺ concentrations, we provided the new relevant data of LiTmF₄: x%Er@LiYF₄ in the revised manuscript, and the spectra have also been improved."

9. The Fig. 2f-i demonstrated the luminescence properties of LiTmF₄:x%Er@LiYF₄ NPs (LiYF₄ shell), but the Supplementary Fig. 8 showed the TEM images of LiTmF₄:x%Er@LiLuF₄ NPs (LiLuY₄ shell). Why?

Response: We are sorry for the carelessness. The sample information of the luminescence spectral data is mislabeled, which are actually LiLuF₄ shells instead of LiYF₄ shells. It also led to confusion about the lifetime results of Tm@Y NPs in the subsequent comments. Also, see our response below.

10. The lifetime of Tm@Y is very confusing. In Fig. 2b, the lifetime of LiTmF₄@LiYF₄ (when x% Tm = 100) is 19.1 ms. But in Fig. 2g, the lifetime of LiTmF₄@LiYF₄ (when x% Er = 0) is 2.5 ms. Which one is real?

Response: Sorry for the carelessness. Both the lifetime of LiTmF₄@LiYF₄ are real, but differences in lifetimes are caused by these two samples being from different sets of groups. Specifically, we did not mention specific differences and the lack of relative TEM results. In these two sets of experiments, the lifetime differs greatly due to the different sizes of the LiTmF₄ core and the different hosts of the shell (one is LiYF₄, and

the other is LiLuF₄).

The specific details are as follows:

1) With the Tm³⁺ doping level increasing in the LiYF₄:xTm% core, the size of the nanoparticles decreases due to the different ionic radii. To ensure that their luminescence is comparable, we controlled the synthesis conditions to make them comparable in size, which led to a relatively larger size of Tm@Y NPs. As a result, the as-synthesized LiTmF₄ core is larger, resulting in a larger size of the core-shell NPs (brighter) and a longer lifetime.

2) As shown in original Fig.2g (incorrect legends, Tm:x%Er@Y), actually, the samples are LiTmF₄-core with LiLuF₄-shell, which can be confirmed by the captions of TEM images in the original supporting information (original Fig. S8). Namely, one is LiYF₄, and the other is LiLuF₄. Moreover, as confirmed by luminescence profiles (Supplementary Fig. 15), the Y host is brighter than the Lu shell, indicating a longer lifetime of the Y shell (minor lattice mismatches and fewer interface defects but stronger emission).

Therefore, it means that the difference in lifetime between the two NPs is mainly due to the nanoparticle size, shell thickness, and the difference in the host.

In the revised manuscript, to ensure the rigour of the article, we supplemented x%Tm@Y NPs (Supplementary Fig.5) and Tm:x%Er@Y NPs (Supplementary Fig.10.) based on a similar size and updated the relevant data. It should be emphasized that this does not affect the conclusion.

11. The authors claimed "Interestingly, the luminescence intensity of Tm-NPs@PEG in aqueous solution decreased much more than it did in cyclohexane (Fig. 3e)". In which window? It is confusing

Response: Sorry for the confusion. Fig 3e in the original manuscript is the data normalized at 1680 nm. The original data is shown in the following Figure. The excitation intensity of Tm-NPs decreased integrally by 9.4 times (calculated at 1680 nm) after PEGylation and dispersed in water. Here, we wanted to investigate solvent effects on the luminescence of Tm-NPs in different bands. It can be seen that when the NPs were dissolved in cyclohexane, the ratio of the intensity at 1738 nm emission to that of 1680 nm emission is 0.37, while the ratio increase to 0.76 when dissolved in water.

Original: Fig.3e

Revision: Fig.3e

12. In Fig. 3e, how much the luminescence intensity of the Tm@Y in aqueous solution decreased compared with in cyclohexane?

Response: Luminescence intensity of different regions is affected differently by the solvent. Compared with the Tm@Y NPs in cyclohexane, the luminescence intensity of Tm@Y NPs in the aqueous solution is decreased by a factor of 9.4 (calculated by the peak emission of 1680 nm) under the same conditions.

13. In Fig. 3, the authors compared the in vitro imaging in the NIR-IIb (Er-NPs) and the NIR-IIc (Tm-NPs) windows. The authors should also compare the in vivo imaging in the NIR-IIb and the NIR-IIc windows in Fig. 4, by using these two probes.

Response: Thanks for the suggestion. To comprehensively evaluate NIR-IIb and c imaging profiles, we performed in vitro and in vivo studies using Er- and Tm-based nanoprobes in the revised manuscript.

For in vivo imaging, the mixture of Er- and Tm- probes were injected into the mouse. This imaging method could compare in vivo NIR-IIb and c imaging in situ. It could be observed from Fig. 4d that NIR-IIc imaging could distinguish tiny vessels with higher imaging resolution, and the SBR reaches 1.56, while that of NIR-IIb is 1.27 (Fig. 4e). These results confirmed the advantage of NIR-IIc imaging.

Fig.4d, Vascular luminescence imaging with Tm- and Er-based NPs at the same concentration in NIR-IIb (1,500 nm LP+1,700 nm SP filters,1500-1700 nm) and NIR-IIc (1,700 nm LP filter) subwindow. **e**, corresponding luminescence intensity statistics of **d**.

14. 1600 nm or 1700 nm LP filter should be used instead of 1500 nm LP filter in Fig. 4 to ensure the NIR images taken in the NIR-IIb/c region of 1600-2100 nm.

Response: Thanks for the suggestion. To ensure that the NIR images were taken in the NIR-IIb/c region, we performed studies using the MCT camera (0.8-2.5 μm) with various filters, including 1600 nm LP, 1700 nm LP, 1500 nm LP+1700 nm SP (1500-1700 nm) and 1750-2250 nm. The details of imaging subwindows collected using the

MCT camera with various filters are as below:

Filters	Transmittance window	Imaging Subwindows	Purpose
1500 nm LP(original)	>1500 nm	NIR-IIb to d	Evaluation of NIR-IIb-c imaging performance of Tm probe (original)
1600 nm LP	>1600 nm	NIR-IIb to d	Evaluation of NIR-IIb-c imaging performance of Tm probe
1700 nm LP	>1700 nm	NIR-IIc to d	Evaluation of NIR-IIc imaging performance of Tm probe
1500 nm LP+1700 nm SP	1500-1700 nm	NIR-IIb	Distinguish between NIR-IIb and c imaging for comparing Er and Tm probe imaging performance
2000-500 nm BP	1750-2250 nm	NIR-IIc	Distinguish between NIR-IIb and c imaging for comparing Er and Tm probe imaging performance

15. The authors claimed "A bright, clear vascular arrangement appeared in the mouse body (Fig. 4a)." I did not see vascular in Fig. 4a. Was the skin removed? If so, the authors need to give more details about this experiment.

Response: Sorry for the confusion. It should be Fig.4b, not Fig.4a, because Fig. 4a is the brightfield image. Fig. 4a is an image of the mouse under a bright field, showing the state of depilation and the modelling state of the venous thrombosis in the neck. In order to better demonstrate the imaging performance in NIR-IIc region, we

supplemented the imaging of the mouse cerebral infarction model (As suggested by reviewer #2) and whole body vascular imaging using the MCT camera. In addition, we provided more details about the imaging experiment in the revised manuscript.

16. The definition of optical window of NIR-I and NIR-II in Fig. 4k is confusing.

Response: Sorry for the confusion. We classified 800 nm as NIR-I and 980 nm, 1,208 nm and 1,530 nm as NIR-II according to the definitions in the literature (doi:10.7150/thno.31332, 10.1021/acsnano.8b02452). Tm(30Er)-NPs nanoprobe can be excited by four wavelengths and emit NIR-IIb/c fluorescence (1,600-2,100 nm). We have made corresponding corrections, as shown in Fig 4i.

Ref.1: Feng, Z., Yu, XM., Jiang, MX. et al. Excretable IR-820 for *in vivo* NIR-II fluorescence cerebrovascular imaging and photothermal therapy of subcutaneous tumor. *Theranostics* 2019; 9(19):5706-5719. <https://doi.org/10.7150/thno.31332>.

Ref.2: Qi, J., Sun, CW., Li, DY. et al. Aggregation-Induced Emission Luminogen with Near-Infrared-II Excitation and Near-Infrared-I Emission for Ultradeep Intravital Two-Photon Microscopy. *ACS Nano* 2018, 12(8):7936-7945. <https://doi.org/10.1021/acsnano.8b02452>.

Original: Fig.4k

Revision: Fig.4i

k	Optical window	Ex (nm)	Em (nm)
	NIR-I	800 980	1600-2100
NIR-II	1208 1530	1600-2100	

i	
Ext. (nm)	Em. (nm)
NIR-I ₈₀₀	
NIR-II ₉₈₀	NIR-IIb/c
NIR-II _{1,208}	(1,600-2,100)
NIR-II _{1,530}	

17. How is the Tm-NPs@PEG circulation in mouse body after intravenous injection?

It is important to know for in vivo imaging.

Response: Thanks for the suggestion, whole-body luminescence imaging of mice was studied, which provides a good representation of the Tm-NPs@PEG circulation *in vivo*, as shown in Fig 4a and Supplementary Fig. 21. An intense NIR-IIc signal allowed clear visualization of the circulatory system 2 min postinjection. The signal decreased with an increase in postinjection time. The liver was clearly seen over time (Figure 4a), indicating that Tm-probe metabolized mainly through hepatobiliary system. Furthermore, we found that Tm-NPs exhibited higher imaging quality and longer half-life (49 min) compared to FDA-approved ICG dye (3 min)[Adv Mater 2021, e2006902. DOI: 10.1002/adma.202006902].

Fig. 4a Real-time vascular luminescence imaging with Tm(02Er)-NPs@PEG in living mice a, at NIR-IIc region.

Supplementary Fig. 21. Blood circulation of Tm-based nanoprobe administrated mice as a function of time.

18. In Supplementary Fig. 5, histograms demonstrating the particle size distribution are needed.

Response: Histogram statistics of particle size are provided in the revised manuscript.

Supplementary Fig. 1. TEM images of LiTmF₄ core and LiTmF₄@LiYF₄ core-shell NPs.

19. Line 149, the "linearly" is not correct.

Response: Thanks for your advice. We changed it to "monotonically".

20. The reference 9 and 14 are the same.

Response: We have corrected it.

21. Check the format of reference 3, 5, 13, 25 and 30.

Response: We checked the format of references throughout in revised manuscript.

Reviewer #1 (Remarks to the Author):

The authors well conducted the imaging experiments using an MCT camera and carefully rechecked the whole manuscript. They did develop excellent emissive probes beyond 1700 nm. It is a novel and meaningful work, which meets the publishing requirements in Nat Commun in my opinion.

However, the claimed imaging advantages might operate only using the specific filters and the specific probes in this work, since the imaging windows were determined both by the filters and the emission ranges of the probes. To absolutely compare the near-infrared regions, it is essential to use probes with even emissions in the specific windows, for example. Thus, I recommend the authors do some necessary discussion about the window advantages in the manuscript before being accepted.

Reviewer #2 (Remarks to the Author):

The authors have revised the manuscript according to the reviewers' comments. Now it is acceptable.

Reviewer #3 (Remarks to the Author):

I have a few more issues that the authors should address:

1) The claim of 'Furthermore, the luminescence quantum yield (1,600-2,100 nm for Tm-NPs) was estimated to be $20.6 \pm 4.1\%$ with 800 nm excitation (Supplementary Fig. 6 and Table1)' seems to be super high, higher than any probes reported among NIR-IIb or IIc probes. Rigorous proof is needed.

2) There are a few of vendors can provide integrating sphere with reflectance window of 250-2500nm. The reviewer still strongly suggests to measure the absolute quantum yield. The absolute quantum yield in aqueous solution should also be measured.

3). What are the laser intensities of 980 nm, 1280 nm and 1530 nm lasers. What's the power/fluence used for each image shown in the paper?

4). The MCT camera for NIR-IIc imaging shows a strong background, which will influence the imaging performance. A comparison between an InGaAs camera and the HgCdTe camera on background and SBR by imaging the same sample in the NIR-IIb window can be added in the Supplementary Information.

5) UV vis NIR absorption curves are missing for the probes. How much light is absorbed at 808 nm and other excitation wavelengths? How does the absorption data comparing to literature (in terms of light absorption by Tm ions)?

6) The authors should give detailed imaging conditions, such as power of laser, illumination fluence and comparison with laser safety limit, exposure times et al., for all the imaging figures in the paper.

Responses to the reviewers' comments (Manuscript number: NCOMMS-22-15435-A for Nature Communications)

Title: Bright Tm³⁺-based downshifting luminescence nanoprobe operating around 1,800 nm for NIR-IIb and c bioimaging

REVIEWER COMMENTS

Reviewer #1 (Remarks to the Author):

The authors well conducted the imaging experiments using an MCT camera and carefully rechecked the whole manuscript. They did develop excellent emissive probes beyond 1700 nm. It is a novel and meaningful work, which meets the publishing requirements in Nat Commun in my opinion.

However, the claimed imaging advantages might operate only using the specific filters and the specific probes in this work, since the imaging windows were determined both by the filters and the emission ranges of the probes. To absolutely compare the near-infrared regions, it is essential to use probes with even emissions in the specific windows, for example. Thus, I recommend the authors do some necessary discussion about the window advantages in the manuscript before being accepted.

Response: We thank the reviewer for the positive evaluation and suggestion on our work. As suggested, we have conducted some necessary discussion in terms of window advantages in the revised manuscript and marked it in blue.

Reviewer #2 (Remarks to the Author):

The authors have revised the manuscript according to the reviewers' comments. Now it is acceptable.

Response: We thank the reviewer for his/her recommendation.

Reviewer #3 (Remarks to the Author):

I have a few more issues that the authors should address:

1) The claim of 'Furthermore, the luminescence quantum yield (1,600-2,100 nm for Tm-NPs) was estimated to be $20.6 \pm 4.1\%$ with 800 nm excitation (Supplementary Fig. 6 and Table1)' seems to be super high, higher than any probes reported among NIR-IIb or IIc probes. Rigorous proof is needed.

Response: Thanks for the reviewer's suggestion. As suggested, we measured the absolute quantum yield of samples. See comment #2 for details.

2) There are a few of vendors can provide integrating sphere with reflectance window of 250-2500nm. The reviewer still strongly suggests to measure the absolute quantum yield. The absolute quantum yield in aqueous solution should also be measured.

Response: Thanks for the reviewer's suggestion. To the best of our ability, we finally got technical support for the QY test from Ocean Insight Company (Shanghai). The measured quantum yield (1,600-2,050 nm) of the nanoparticles is 14.16% and 16.13% for Tm-NPs and Tm(O₂Er)-NPs, respectively, by using an integrating sphere with the spectrometers. However, this QY measurement system cannot provide for solution medium. Therefore, the quantum yield of probes in the aqueous cannot be obtained; however, the luminescent properties of the probes can still be reflected by the powder samples.

The method and the configuration were supplemented in the SI.

The powder samples were excited by an 800 nm laser (CNI, Changchun) at $4\text{W}/\text{cm}^2$. An integrating sphere (Labshere, 3.3 inches) was used to spread the multiple light reflections over the entire sphere surface. The spectrometer collected the outcome lights, including excitation light and emission light of samples. The attenuator and inline filter holder (including a 900 nm LP filter) were used during the test. First, calibrate the dual channel spectrometer (QE PRO, 783-1,027 nm, Slit 50 μm and NQ512, 895-2,131 nm, Slit 200 μm) with the calibration light source (HL-3-INT-Cal,

~2,500 nm) to obtain the radiation flux curve. Next, adjust the appropriate integration time for the two spectrometers to enable them to obtain enough effective signals, and then update the background spectrum immediately. Next, spectrometer QE was used to obtain excitation light radiation flux data. Finally, we used spectrometer NQ to obtain the emission light radiation flux data. The software (Ocean QY 2.02) automatically calculated the quantum yield (QY) using the following formula: $QY = \text{photons emitted} / \text{photons absorbed}$.

3). What are the laser intensities of 980 nm, 1280 nm and 1530 nm lasers. What's the power/fluence used for each image shown in the paper?

Response: Specific laser intensity conditions have been provided in the experimental section in the previous version. These three lasers had the same power as the 800 nm at 120 mW/cm^2 for four wavelengths of excitation imaging. Such as in the caption of Figures 3 and 4 and the experimental section. To increase the paper's readability, we further provide the experimental conditions in the appropriate locations accordingly.

4). The MCT camera for NIR-IIc imaging shows a strong background, which will influence the imaging performance. A comparison between an InGaAs camera and the HgCdTe camera on background and SBR by imaging the same sample in the NIR-IIb window can be added in the Supplementary Information.

Response: Thanks for the suggestion. In fact, the imaging quality is related to the camera's performance, spectra response range, integration time, and many other factors. As suggested, we supplemented the NIR-IIb imaging with InGaAs (SD640) and MCT camera on the background and SBR by imaging the same sample under the same irradiation conditions. The results were provided in the Supplementary Information.

Considering the QE of the InGaAs camera at the 1,600-1,700 nm range, we adopted a higher excitation power (using an 800 nm laser excitation and luminescence detection in the 1,600-1,700 nm range). Whole-body blood vessel imaging of the same mouse but different cameras is performed with the same excitation power (200 mW/cm^2 within the safety limit) and exposure time (50 ms). However, the difference is that the MCT camera operates in a low gain, while the InGaAs camera is used in a high gain mode.

As shown in Supplementary Fig. 22, the SBR of the NIR-IIb images using the MCT camera is similar to that of InGaAs imaging (~ 2.2). Further, it can be seen that the background noise of MCT is lower when the same signal threshold is set. Therefore, it suggests that Tm-based probes are excellent NIR-IIb imaging probes, which can also image well on InGaAs cameras.

Supplementary Fig. 22. a. Comparison of NIR-IIb luminescence imaging of blood vessels in the same mouse, treated with Tm02Er nanoprobe under 800 nm excitation (200 mW/cm^2) with 1,600 nm LP and 1,700 nm SP filters. Corresponding cross-sectional luminescence intensity profiles along blue lines. Gaussian fits are shown in the red line. **b.** The blood vessel imaging with InGaAs and MCT camera of the original image, target signal and noise (analyzed with Matable R2021a).

5) UV vis NIR absorption curves are missing for the probes. How much light is absorbed at 808 nm and other excitation wavelengths? How does the absorption data comparing to literature (in terms of light absorption by Tm ions)?

Response: Thanks for the suggestion. We supplemented the absorption spectra of the probes. The absorption of Tm-NPs is shown in Figure S8 (absorption cross-section: $8.25 \times 10^{-22} \text{ cm}^2$ at 800 nm and $1.5 \times 10^{-21} \text{ cm}^2$ at 1,208 nm). Tm(30Er)-NPs can be excited by four wavelengths due to Er³⁺ increases the absorption at 980 and 1,530 nm (absorption cross-section: $9.5 \times 10^{-22} \text{ cm}^2$ at 800 nm, $1.075 \times 10^{-21} \text{ cm}^2$ at 980 nm, $1.75 \times 10^{-21} \text{ cm}^2$ at 1,208 nm and $2.12 \times 10^{-21} \text{ cm}^2$ at 1,530 nm). Compared with the traditional YbTm co-doped system, Tm(30Er)-NPs exhibit more abundant absorption wavelengths in the NIR region. Compared with the absorption cross sections of Tm ions in the literature, they are on the same order of magnitude. (Refs: Energy-Looping Nanoparticles: Harnessing Excited-State Absorption for Deep-Tissue Imaging, ACS Nano 2016, 10, 9, 8423–8433; Growth and spectroscopy of Tm³⁺,Ho³⁺ co-doped LuYO₃ single crystal for 2.1 μm laser, Journal of Luminescence 234 (2021) 117951)

Supplementary Fig. 8a. Absorption spectra of NaYF₄:25%Yb,0.5%Tm, Tm(30Er)-NPs and Tm-NPs in UV-VIS-NIR region.

6) The authors should give detailed imaging conditions, such as power of laser,

illumination fluence and comparison with laser safety limit, exposure times et al., for all the imaging figures in the paper.

Response: We further supplemented necessary imaging conditions accordingly.

For instance, In vivo NIR-II imaging was carried out using the MCT detector upon 800 nm excitation (100 mW/cm^2) with an exposure time of 100 ms. In addition, imaging was recorded with multiwavelength excitation at 800 nm, 980 nm, 1,208 nm, and 1,530 nm ($\sim 120 \text{ mW/cm}^2$ for each). The exposure time of all images was 100 ms. A comparison between an InGaAs (SD 640, Tekwin) and MCT camera on background and SBR by vascular imaging under 800 nm light excitation (200 mW/cm^2) with an exposure time of 50 ms.

Compared with the maximum permissible exposure (MPE) for skin exposure to a laser for wavelengths at 800 nm (0.33 W/cm^2), 980 nm (0.72 W/cm^2), 1,208 nm (1 W/cm^2) and 1,530 nm (1 J/cm^2) (American National Standard for safe use of lasers, ANSI Z136.1-2014), our imaging conditions for in vivo imaging are lower (within the safety limit).

Reviewer #1 (Remarks to the Author):

The authors have addressed all my concerns, and the MS can be published in the current form.

Reviewer #3 (Remarks to the Author):

I now recommend acceptance of the manuscript. It is good the authors removed the claim of ultra high QY.

A point-by-point response to the reviewers' comments

REVIEWERS' COMMENTS

Reviewer #1 (Remarks to the Author):

The authors have addressed all my concerns, and the MS can be published in the current form.

Response: We thank the reviewer for his/her recommendation.

Reviewer #3 (Remarks to the Author):

I now recommend acceptance of the manuscript. It is good the authors removed the claim of ultra high QY.

Response: We thank the reviewer for his/her recommendation and the comment on our removing the claim of ultra-high QY.